# Identifying immunologically-vulnerable regions of the HCV E2 glycoprotein and broadly neutralizing antibodies that target them

Ahmed A. Quadeer [1], Raymond H.Y. Louie [1,2,3,4] & Matthew R. McKay [1,5]

Isolation of broadly neutralizing human monoclonal antibodies (HmAbs) targeting the E2 glycoprotein of Hepatitis C virus (HCV) has sparked hope for effective vaccine development. Nonetheless, escape mutations have been reported. Ideally, a potent vaccine should elicit HmAbs that target regions of E2 that are most difficult to escape. Here, aimed at addressing this challenge, we develop a predictive in-silico evolutionary model for E2 that identifies one such region, a specific antigenic domain, making it an attractive target for a robust antibody response. Specific broadly neutralizing HmAbs that appear difficult to escape from are also identified. By providing a framework for identifying vulnerable regions of E2 and for assessing the potency of specific antibodies, our results can aid the rational design of an effective prophylactic HCV vaccine.

---

[1] Department of Electronic and Computer Engineering, The Hong Kong University of Science and Technology, Clear Water Bay, Hong Kong, China. [2] Institute for Advanced Study, The Hong Kong University of Science and Technology, Clear Water Bay, Hong Kong, China. [3] The Kirby Institute, University of New South Wales, Sydney 2052, Australia. [4] School of Medical Sciences, University of New South Wales, Sydney 2052, Australia. [5] Department of Chemical and Biological Engineering, The Hong Kong University of Science and Technology, Clear Water Bay, Hong Kong, China. Correspondence and requests for materials should be addressed to M.R.M. (email: m.mckay@ust.hk)

HCV infection currently affects around 71 million people worldwide[1], with this number increasing by 3–4 million each year. Around 20–30% of infections are asymptomatic and resolve within 6 months, while the remaining persistent infections often lead to chronic hepatitis, fibrosis, cirrhosis, and liver cancer[2]. Although notable treatment therapies based on antiviral agents have been recently developed, these are generally expensive and have limited efficacy due to the appearance of drug resistance mutations[3]. There is currently a need for a potent HCV vaccine.

A confounding factor thwarting the development of a HCV vaccine is the high replication rate ($\sim 10^{12}$ copies per day[4]) and mutation rate ($\sim 10^{-4}$ mutations per nucleotide per replication cycle[5,6]), which allows the virus to escape from human immune responses. However, the recent discovery of HmAbs capable of neutralizing numerous HCV isolates, referred to as broadly neutralizing HmAbs[7–9], and the association of their early appearance with spontaneous viral clearance[10,11] has raised hope for a potent prophylactic HCV vaccine. The major target of these neutralizing HmAbs is the envelope glycoprotein 2 (E2), the most exposed part of the virus which directly interacts with the cellular receptors during viral entry[12,13]. However, the neutralization breadth of these antibodies has been measured against only a few HCV isolates, including the representative isolates of the six HCV genotypes, due largely to the limited availability of diverse replication-competent chimeric HCV strains[14]. Moreover, escape mutations from these broadly neutralizing HmAbs have been reported in multiple experimental studies[15–18], pointing to potential limitations in their efficacy.

An important challenge in the design of a potent prophylactic HCV vaccine is to elicit antibodies that target regions of E2 for which individual mutations needed to escape the associated immune pressure carry a high fitness cost to the virus. This requires a systematic characterization of the fitness landscape, a mapping from the amino acid sequence of a viral strain to a number which quantifies its ability to assemble and propagate infection. The fitness landscape of a highly variable protein like E2 is seemingly complicated, being characterized not only by effects of point mutations at individual residues, but also by interactions (e.g., compensatory or antagonistic) between mutations at multiple residues[19–21]. Experimentally determining such a complex fitness landscape is infeasible as it would require a prohibitively large number of fitness experiments. Knowledge of the fitness landscape of E2 alone, however, is not sufficient to predict viral escape from antibody response. This is because the escape process is mediated through complex (fitness-dependent) stochastic dynamics which involve host–virus interaction, competition between different strains in the evolving virus quasispecies, etc.

To address these issues, we employ a recently proposed efficient computational method to infer an in silico model for the fitness landscape of the HCV E2 protein using available sequences for genotype 1a (one of the most prevalent HCV genotypes worldwide[22]). We validate the inferred model by comparing with numerous experimental data and demonstrating meaningful predictions. Then, we integrate the fitness landscape into a stochastic population genetics model of in-host viral evolution, which we employ to quantify the average time to escape from antibody responses targeting any specific residue in E2. The evolutionary model is validated by comparing with experimental and clinical data. We study the escape time associated with mutations at residues in each of the five known antigenic domains of E2, which reveals one particular domain, namely antigenic domain C, comprising residues at which mutations take predominantly longer time to escape. By using binding information of known broadly neutralizing HmAbs determined experimentally, we also study the effectiveness of these antibodies in neutralizing diverse viral strains. Our analysis suggests that mutations at binding residues of many broadly neutralizing HmAbs are associated with relatively short escape times, pointing to their limited neutralization breadth even within genotype 1a isolates. However, we also discover HmAbs which we predict to be relatively escape-resistant; meaning that mutations at binding residues of these HmAbs take longer time to escape as compared to other HmAbs. This, in turn, points to their potentially enhanced ability to neutralize diverse genotype 1a isolates. The results we report here can aid the rational design of potent HCV vaccines and associated protocols that solicit antibody responses specifically directed toward vulnerable regions of E2.

## Results

**E2 fitness landscape inference and validation.** We inferred an in-silico model for the fitness landscape of E2 using the sequence data available for genotype 1a (Methods). This involved obtaining a prevalence landscape—an estimate of the probability of observing a particular sequence among naturally occurring viral populations—using a maximum entropy (least-biased) probabilistic model[23]. For an arbitrary sequence $\mathbf{x} = [x_1, x_2, \ldots, x_L]$, the model assigns the probability

$$p_{\mathbf{h},\mathbf{J}}(\mathbf{x}) = \frac{e^{-E_{\mathbf{h},\mathbf{J}}(\mathbf{x})}}{Z}, \text{ with } E_{\mathbf{h},\mathbf{J}}(\mathbf{x}) = \sum_{i=1}^{L} h_i(x_i) + \sum_{i=1}^{L} \sum_{j=i+1}^{L} J_{ij}\left(x_i, x_j\right),$$
(1)

where, following the language of statistical physics, $\mathbf{h}$ denotes the set of all fields (representing the effects of point mutations at individual residues), whereas $\mathbf{J}$ denotes the set of all couplings (representing the interactions between mutations at different residues). In addition, $Z$ is a normalization factor while $E_{\mathbf{h},\mathbf{J}}(\mathbf{x})$ is referred to as the energy of strain $\mathbf{x}$. The field and coupling parameters are chosen such that the single and double mutant probabilities of the model match those observed in the multiple sequence alignment (MSA) (Methods). Such maximum entropy models have been used previously for inferring the fitness landscape of the polymerase protein (NS5B) of HCV[24], as well as for several HIV proteins[25–27]. The E2 protein of HCV has much higher mutational diversity compared with all other HCV proteins, and consequently the number of model parameters for E2 is very large (Supplementary Fig. 1). This makes the task of inferring these parameters challenging. Here, to address this challenge, we inferred the model parameters using an efficient computational approach that we introduced previously to infer the complicated fitness landscape of the HIV envelope protein[28].

The single and double mutant probabilities obtained using the inferred model matched well with those of the MSA (Fig. 1a, b), confirming the accuracy of the maximum entropy model fit. Furthermore, although not involved in the model training procedure, additional statistics of the MSA—including the triple mutant probabilities (Fig. 1c) and the distribution of the number of mutations (Fig. 1d)—were also accurately captured by the inferred model.

For the polymerase protein of HCV[24] and multiple HIV proteins[25–28], as well as for bacterial proteins[29], tests against in vitro fitness measurements have demonstrated that the prevalence landscape, inferred using maximum-entropy-based models, is a good representative of the underlying fitness landscape. To assess this correspondence for E2, we compared fitness predictions based on the inferred landscape with in vitro infectivity measurements compiled from the literature[16,30–39]. These include a total of 92 measurements consisting of sequences differing from the wild-type (H77) sequence by between 1 and 30

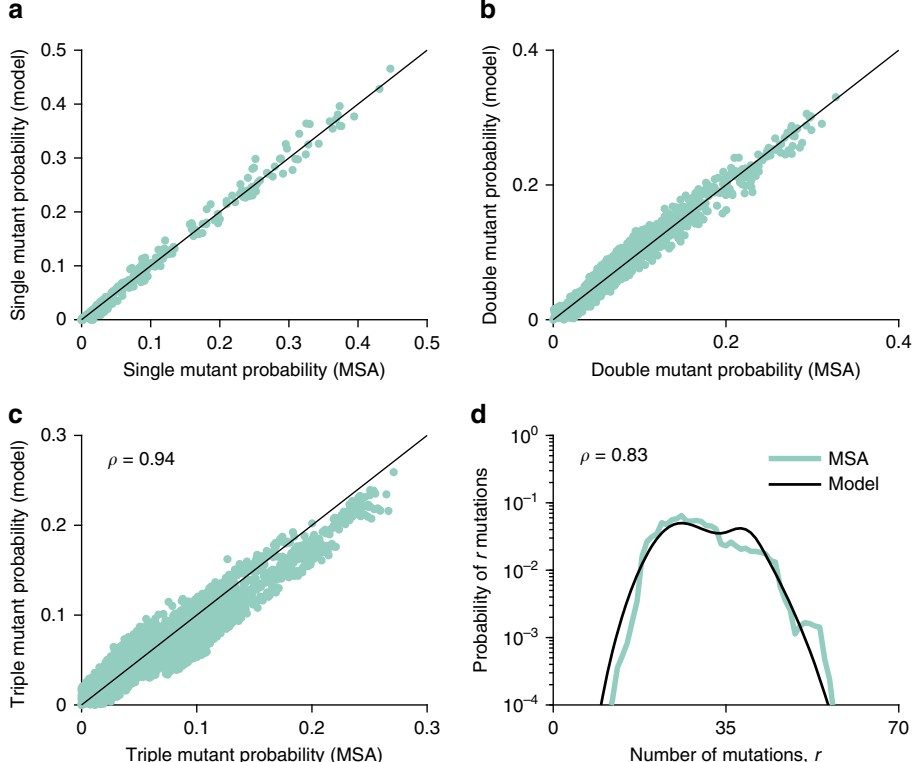

**Fig. 1** Statistical validation of the inferred E2 landscape. The inferred model accurately reproduces the statistics of the observed E2 sequences. Scatter plots of the data statistics, single mutant (**a**) and double mutant probabilities (**b**), used to train the model. Results for the data statistics, the triple mutant probabilities (**c**) and the distribution of the number of mutations (**d**), predicted by the inferred model. A strong Pearson correlation of $\rho = 0.94$ and $\rho = 0.83$ is observed between the MSA and model probabilities, respectively

mutations. We employed the energy of a sequence computed using the inferred model (Eq. (1)) as a metric of fitness, which is inversely related to its prevalence. Therefore, a sequence with low energy is considered to have high fitness and vice versa. A strong negative correlation (Methods) $\bar{r} = -0.72$, was observed between model energies and experimental fitness measurements (Fig. 2a), corroborating that the inferred E2 prevalence landscape indeed serves as a meaningful proxy for the respective intrinsic fitness landscape.

In addition to experimental fitness measurements, we also tested our landscape's ability to predict a specific non-epitope compensatory mutation Q444R, reported to be associated with the escape mutation N417S that conferred resistance to the HmAb HCV1 (binding residues 412–423) in a chimpanzee model[17]. The compensatory effect of Q444R for N417S was confirmed by engineering these mutations in the H77 virus E2 glycoprotein in vitro. To test whether our landscape can capture this effect, we analyzed the change in energy observed by making all possible mutations in the H77 strain carrying the N417S mutant (H77$_{\text{N417S}}$). Mutations yielding a negative change in energy imply an increase in fitness, whereas those yielding a positive energy change would imply a fitness decrease. Quite strikingly, our model predicted four mutations at residue 444 (Q444H, Q444V, Q444Y, and Q444R) to be associated with the most negative change in energy among all mutations in the H77$_{\text{N417S}}$ strain (Fig. 2b), which is consistent with the reported compensatory role of mutation at this residue for the escape mutation N417S. All fitness-improving single mutation variants of the H77$_{\text{N417S}}$ mutant strain are listed in Supplementary Table 1 (for comparison, the same is also shown for the H77 strain, indicating that these fitness-improving variants are not specific to the H77$_{\text{N417S}}$ mutant strain).

**Relative escape time prediction and validation**. The above experimental validations provide confidence in our model's predictive power of viral fitness. However, in order to predict escape from antibodies, one should account for the complex stochastic dynamics involved with in-host viral evolution. This includes, for example, effects of host–virus interaction, multiple possible pathways that may be used by the virus to evade host immune response, and sequence heterogeneity of the evolving virus quasispecies. To incorporate these stochastic effects, following a similar approach to recent work on HIV[40], we incorporated the predicted E2 fitness landscape into a Wright–Fisher-like population genetics model[41] for simulating the in-host viral evolution. The parameters of this model, such as mutation rate and effective population size, were set according to known values for HCV, while the inferred E2 fitness landscape was used to determine the probability of survival of each sequence in the population from one generation to the next (see Methods for details).

For each E2 residue, we used this simulation model to predict the time for an escape mutation to reach a majority in the population. Specifically, for a given residue, we started the simulation with an initial homogeneous population comprising a randomly selected MSA sequence with the consensus amino acid at that residue. Antibody pressure was modeled such that any sequence in the population with consensus amino acid at that residue was penalized by a fixed reduction in fitness, chosen large enough to confer a selective advantage to sequences with a mutant at the residue. This evolutionary simulation continued until the frequency of sequences in the population with a mutant at the residue was sufficiently high (reaching above 0.5), at which time escape was said to have occurred. The number of generations for escape was recorded, and this procedure was

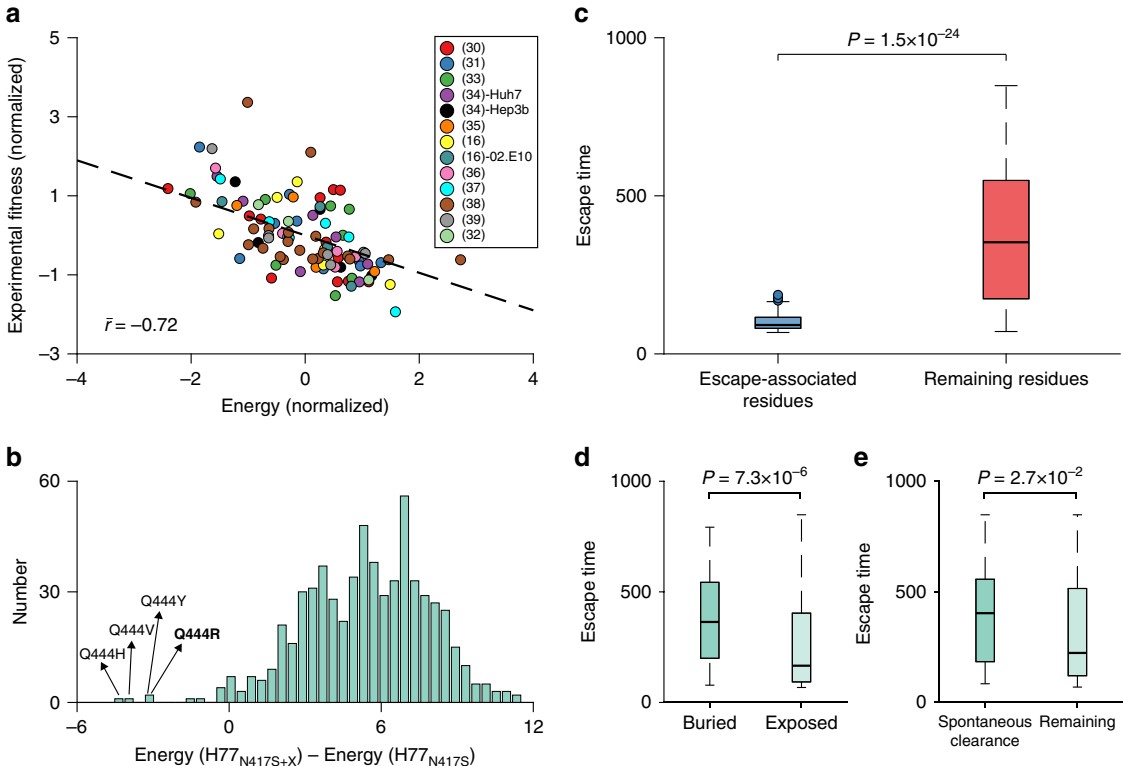

**Fig. 2** Biological validation of inferred E2 landscape and escape time metric. **a** Normalized experimental fitness measurements correlate highly with the energy computed from the inferred landscape. References for fitness (infectivity) measurements (Supplementary Data 3) are shown in the legend. Fitness measurements were reported using two different reference strains (H77 and 02.E10) in ref. [16] and two cell lines (Huh7 and Hep3b) in ref. [34]. Normalization for energy and fitness measurements is performed by subtracting the mean from each data set and dividing by its standard deviation. The correlation $\bar{r}$ is the mean of the Spearman correlation between the predicted energy and the fitness data reported in each reference, weighted with the respective number of measurements (see Supplementary Fig. 2 for plots of individual experiments). The black dashed line represents a linear least-squares fit. **b** Histogram of the change in energy observed by all single mutations X in the H77 strain carrying the N417S mutant. Energy(H77$_{N417S}$) and Energy(H77$_{N417S+X}$) is the predicted energy for the H77 strain carrying the N417S mutant and for the H77 strain carrying both the N417S mutant and an additional single mutation X, respectively. Mutations with the most negative change in energy are labeled on the plots. The Q444R mutation reported[17] to be compensating for N417S is shown in bold. **c** Comparison of the distribution of escape times ($t_e^i$) associated with experimentally/clinically observed escape mutations from multiple HmAbs (Supplementary Table 2) and that of the remaining protein residues. In each box plot, the bold horizontal line indicates the median, the edges of the box represent the first and third quartiles, and whiskers extend to span a 1.5 interquartile range from the edges. **d** Comparison of the distribution of escape times associated with the mutations on buried and exposed residues. The residues are classified as buried and exposed using the recently determined E2 crystal structure[43] (PDB ID 4MWF). **e** Comparison of the distribution of escape times associated with mutations in the T lymphocyte epitopes (Supplementary Table 3) targeted by spontaneous-clearance-associated HLA alleles[46,48,49] and those targeted by other alleles. All reported P values are calculated using the one-sided Mann–Whitney test

repeated many times for the same initial strain, as well as for multiple initial strains (Methods). The mean times over all iterations, termed as escape time, were computed and recorded for every E2 residue (Supplementary Data 1). As the specific values of the escape times depend on the parameters selected for the evolutionary model (see Methods), these should be most meaningfully interpreted in a comparative sense, providing a classification of the relative ease of escape across residues.

We validated the predictions of the evolutionary simulations against multiple experimental and clinical data. First, we tested our model for known escape mutations from multiple E2-specific HmAbs[15–18] (Fig. 2c). One would assume that such escape mutations would generally be associated with lower escape times than mutations at other residues in E2. These escape-associated residues include mutations from specific antibodies that were either determined experimentally or observed over time in chronically infected hosts (listed in Supplementary Table 2). Our results demonstrated that these escape mutations were indeed associated with lower values of escape time ($P = 1.5 \times 10^{-24}$, Mann–Whitney test), as expected.

We further validated our evolutionary model by examining the escape times associated with mutating the buried and exposed (surface) residues in E2. The buried residues that form the protein core are likely to be crucial for stability[42], and thus mutations at these residues are generally expected to incur a higher fitness cost than mutations at the exposed residues. As such, one would anticipate that escape times would be typically longer for these residues, if they were to be targeted by the immune system. While the buried residues may generally not often be accessible to antibodies, studying escape times associated with mutating them is meaningful, since multiple experimentally identified antibody binding residues have been found to be nonexposed[38], and these are also generally accessible to the cellular arm (comprising T lymphocytes) of the adaptive immune system. We identified the buried and exposed residues in E2 from the recently-determined truncated crystal structure of the E2 core domain[43] using the standard relative solvent accessibility (RSA) metric (Methods). The hypervariable region 1 (HVR1), for which the crystal structure is not yet resolved, was included in the set of exposed residues, as it is known to be the most immunogenic region of

E2[44]. Comparing the escape times associated with the two sets of residues revealed that the mutations at buried residues were indeed predicted to have longer escape times ($P = 7.3 \times 10^{-6}$, Mann–Whitney test) (Fig. 2d).

While our main focus is on antibody-based immune responses, we assessed the predictions of our model for residues involved with escape from cytotoxic T lymphocyte (CTL) responses. Although E2 is a major target of antibodies, multiple experimental studies have reported that it is also subjected to a CTL response[45]. In fact, clinical studies have suggested the HLA-B57 allele to be associated with spontaneous viral clearance due to targeting of a specific CTL E2 epitope[46]. We compared the escape times associated with mutating the residues in all E2-specific epitopes available in the HIV Molecular Immunology Database[47] (listed in Supplementary Table 3). This study suggested a lack of residues with shorter escape times in epitopes targeted by the HLA alleles (HLA-B57, HLA-B51, and HLA-A3) associated with spontaneous viral clearance[46,48,49] ($P = 2.7 \times 10^{-2}$, Mann–Whitney test), corroborating the possible role of targeting these epitopes in combating HCV (Fig. 2e).

**Identification of antigenic domains resistant to escape.** Having provided numerous sources of validation of our model's predictive power, we applied it to identify exposed regions in E2 that may be targeted by HmAbs for stimulating a robust antibody response. We did this by superimposing the escape time associated with mutations at each residue of E2 as a heat map on the protein structure (Fig. 3a). A reasonable proportion of red colored residues in this map suggested an abundance of mutations on the E2 surface associated with relatively longer escape times. This is encouraging from an immunological perspective as it raises the possibility of rationally designing antibodies[50] that specifically target the exposed E2 residues which appear most difficult to

escape from, and thereby for eliciting a robust and potentially effective immune response.

We further studied the escape times associated with regions in E2 that are known to be targeted by HmAbs. Specifically, we analyzed the following three regions: (i) HVR1 which plays an important role during viral entry by interacting with the scavenger receptor class B type 1 (SR-B1) receptor[13]; (ii) the co-receptor CD81 binding sites (CD81bs) critical for viral entry[12]; and (iii) the E2 antigenic sites. Different terminologies have been used in the literature for defining these antigenic sites in E2, e.g., antigenic domains[36,38] A–E, antigenic regions[7,9] 1–5, and epitopes[51] I–III. However, these definitions of antigenic sites are largely overlapping (e.g., antigenic domain B and antigenic region 3 share multiple residues, and antigenic domain E and epitope I comprise the same residues). Here, we focused on the five E2 antigenic domains A–E.

We compared the relative ease of escaping immune pressure directed at each domain using two metrics (see Methods for details): (i) The minimum escape time across all residues in a domain, and (ii) the number of residues in a domain having escape times less than or equal to a (small) threshold number of generations. The latter quantity provides a measure of the abundance of residues in the domain that appear to be associated with low escape times. In addition, to get an idea of what may reasonably constitute an easy escape in terms of our computed escape times, we designed a classifier using the information available in the literature for the experimentally/clinically-observed escape mutations from E2-specific HmAbs (Supplementary Table 2). Specifically, a classifier designed using the residues with observed escape mutations as true positive control values and assuming all remaining residues as true negative values achieved an area under the receiver operating characteristic curve

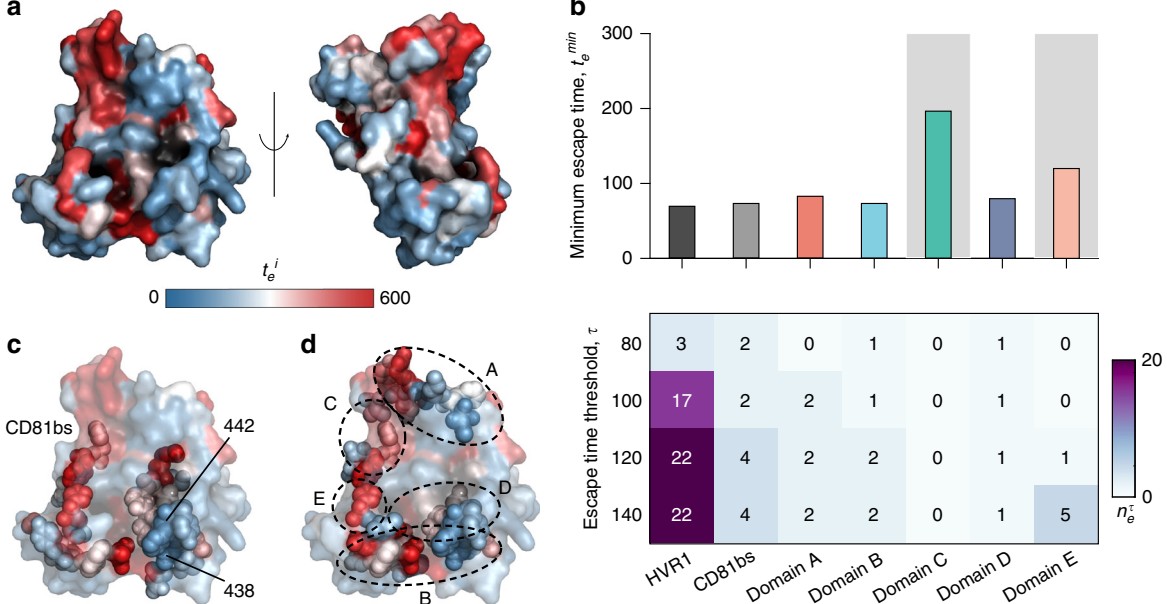

**Fig. 3** Superimposing relative escape times on the available E2 crystal structure reveals difficult to escape regions for antibody targeting. **a** Residues incurring high escape times upon mutation (large values of $t_e^i$) are shown in red and those associated with low escape times (small values of $t_e^i$) are shown in blue. The crystal structure[43] was obtained from the protein data bank with PDB ID 4MWF. Left and right panels show the front and side views of E2, respectively. **b** Analysis of minimum escape time across all residues $t_e^{min}$ (top panel) and number of residues with relatively short escape times $n_e^\tau$ (bottom panel) in E2 regions that are targeted by antibodies. Among all domains accessible to antibodies, antigenic domains C and E (shaded) have $t_e^{min} > \zeta$ (102 generations) and are thus predicted to be difficult to escape. Domain C, in particular, is predicted to be the most escape resistant. **c** Escape times superimposed on the crystal structure with CD81 binding sites highlighted as spheres. Specific residues (438 and 442) associated with low escape times in CD81 binding sites are also highlighted on the figure. **d** Escape times superimposed on the crystal structure with antigenic domains A–E highlighted as spheres. Approximate location of specific antigenic domains is shown by dotted ellipses. HVR1 and a large part of domain E is not shown as the involved residues are not resolved in the structure

(AUC) of 0.93 (Supplementary Fig. 3a). Due to the imbalanced size of the two classes, i.e., the number of residues with observed escape mutations being around 1/5th that of the remaining E2 residues, we calculated an optimal escape time cut-off value $\zeta$ based on the F1 score and the Matthews correlation coefficient—two widely used metrics to gauge the performance of binary classifiers with imbalanced dataset. Both metrics returned the same value of $\zeta \sim 100$ generations (Supplementary Fig. 3b). This suggests that, based on the available information about the escape mutations from E2-specific HmAbs, any residue which our model ascribes an escape time of less than $\zeta$ generations may be considered potentially easy to escape and vice versa. The choice of $\zeta \sim 100$ is a statistically reasonable cut-off; though, if one wished to further decrease the chance of incorrectly classifying a residue as difficult to escape, a larger value of $\zeta$ could also be considered.

Among the studied domains, our model associated HVR1 with the lowest escape time, which was much less than $\zeta$ (Fig. 3b, top panel), and the largest number of residues associated with low-escape times (Fig. 3b, bottom panel). This is indicative of the relative ease by which the virus may escape HVR1-specific antibodies, which is consistent with the fact that these are generally known to be strain-specific and fail to neutralize viral variants generated by the high variability of this region[52]. Interestingly, the minimum escape time for CD81bs was predicted to be close to that of HVR1 (Fig. 3b, top panel). This is in contrast to its importance in viral entry and in eliciting neutralizing antibody response[53]. Further investigation revealed that this surprising result was due to two residues with relatively low escape times (residues 438 and 442) in the CD81bs (Fig. 3c). Compared to other domains (Fig. 3b, bottom panel), CD81bs also contained a relative abundance of residues associated with low escape times. From our model predictions, the minimum escape time for antigenic domains A, B, and D appeared to be similar to HVR1 (Fig. 3b, top panel), implying the possibility of reasonably fast viral escape from antibodies targeting these regions. In contrast, domains C and E seemed to be comparatively difficult to escape, as the associated minimum escape time was relatively larger than all other domains and also larger than $\zeta$ (Fig. 3b, top panel). However, domain E still seemed to comprise multiple residues with low escape times (Fig. 3b, bottom panel). Such escape mutations (at residues 415 and 417) in domain E have indeed been reported in a controlled experiment on chimpanzees[17] and in humans during a phase 2 clinical trial[54,55]. Overall, with the largest minimum escape time and with no identified residues with low escape times, domain C stands out as a region of E2 that seems most resistant to antibody escape (Fig. 3b, d). This suggests the potential importance of HmAbs which specifically target this domain for HCV vaccine development.

**Escape resistance of HmAbs.** By employing our model together with experimental data regarding antibody binding, we assessed the efficacy of known broadly neutralizing HmAbs. The most effective antibodies are expected to be those which bind primarily to E2 residues that are associated with relatively large escape time, and thereby restrict easy escape. The binding residues of broadly neutralizing HmAbs isolated from HCV-infected patients have been defined experimentally using various methods. These include structural studies involving binding of antibodies to short E2 peptides[36,56] or E2 core domain[43], global E2 alanine scanning mutagenesis[38], or selective alanine scanning mutagenesis of limited sets of E2 residues (listed in Supplementary Table 4). Here we focused on the antibody binding sites reported using both global and selective alanine scanning mutagenesis. Note that a particular antibody may bind to a subset of residues in an antigenic domain, multiple additional residues surrounding an antigenic domain, or to multiple antigenic domains.

Considering first the global alanine scanning method, which provides the most comprehensive information about the residues critical for antibody binding, we defined the set of binding residues for each HmAb as the residues which when substituted by alanine (or glycine in case of residues with alanine as wild-type) cause a reduction of binding relative to the wild-type (RB) to less than or equal to 20% (Fig. 4a). Note that in addition to exposed residues, these reported binding residues involve a large proportion of buried residues that presumably impact antibody recognition by conformational changes in the protein or through effects on local or global protein stability[38].

Our model predicted that numerous binding residues of all domain A specific HmAbs have relatively low escape times, much smaller than $\zeta$ (Fig. 4a). This is in line with the high flexibility of this domain and the known nonneutralizing nature of the associated antibodies[57]. In contrast, our analysis (Fig. 4a) revealed at least one HmAb in all other domains with binding residues having comparatively large escape times (greater than $\zeta$). Specifically, these included domain B HmAb HC-1, domain C HmAbs CBH-23 and CBH-7, domain D HmAb HC84-20, and domain E HmAb HC33-1. Of these, domain E HmAb HC33-1 was associated with the largest escape time (~300 generations), and thus appeared the most difficult to escape (Fig. 4a). The number of HmAbs predicted to be relatively escape-resistant reduced to two (CBH-7 and HC33-1) when a relaxed threshold of RB $\leq$ 40% was used for defining binding residues of these HmAbs (Supplementary Fig. 4). Taken together, our model suggests that many existing broadly neutralizing HmAbs may permit relatively easy escape, even within genotype 1a. It also identifies that HmAbs HC-1, CBH-23, CBH-7, HC84-20, and HC33-1 (Fig. 4b) as being comparatively more difficult to escape, promoting these as candidates in the design of a potentially effective vaccine against HCV.

We additionally investigated the escape times associated with 20 other HmAbs for which the binding residues were determined by selective alanine scanning mutagenesis (Supplementary Table 4). Our model predicted half of these HmAbs with escape time greater than $\zeta$ and thus these appear to be relatively difficult to escape (Supplementary Fig. 5). Of particular interest are the HmAbs specific to antigenic region 3 (AR3A and AR3B) which were previously shown to prevent HCV infection in humanized mice[58].

## Discussion

The association of spontaneous viral clearance with the early appearance of a broad neutralizing antibody response suggests that a rational vaccine design eliciting a targeted escape-resistant antibody response is a promising way forward for an effective HCV vaccine. However, identifying escape-resistant antigenic regions and antibodies targeting such regions experimentally is challenging. To address this problem in a systematic way, we first inferred a model for the fitness landscape of the E2 protein from available sequence data, considering the effects of point mutations as well as interactions between mutations at different residues. Predictions from this model showed consistency with experimental data. We used this fitness information to develop an evolutionary model for computing time to escape antibody pressure directed toward any E2 residue, which thereby enabled us to predict the most difficult to escape regions in E2, as well as to categorize the available HmAbs according to their ease of escape through mutation of their binding residues. We identified specific HmAbs that may serve as potential vaccine components, due to their relative resilience to escape.

In regards to the inference of the fitness landscape, which was trained based on population-level sequence data, the high correlation observed between prevalence and fitness of E2 (Fig. 2a) is not obvious, particularly given the complex evolutionary dynamics of this

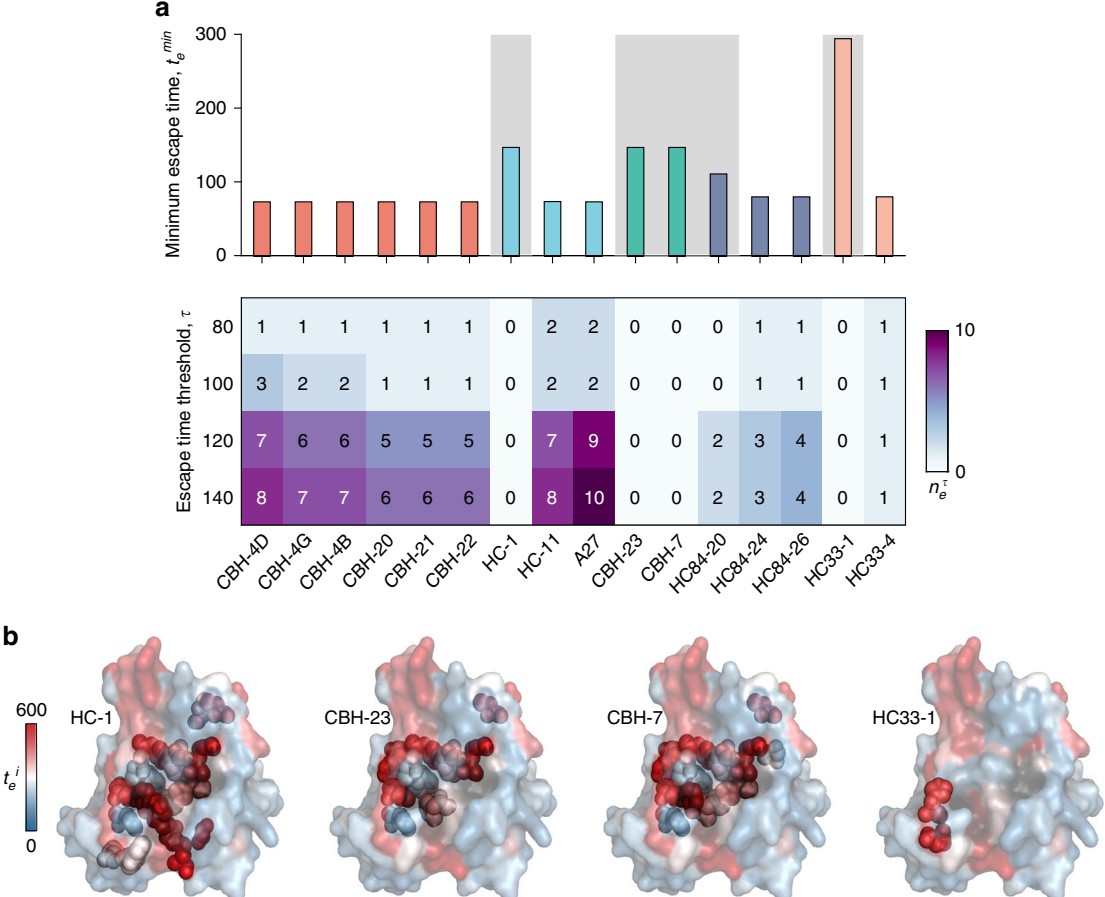

**Fig. 4** Analysis of relative escape times associated with mutating the binding residues of known HmAbs identifies potential difficult to escape HmAbs. **a** Minimum escape time across all residues $t_e^{min}$ (top panel) and number of residues with relatively short escape times $n_e^\tau$ (bottom panel) in the binding residues of 16 HmAbs, determined using global alanine scanning[38]. The binding residues of the HmAbs are defined as the mutations that resulted in reduction of binding with respect to the wild-type to ≤ 20%. HmAbs are colored according to the antigenic domains using the scheme in Fig. 3b. The background of the HmAbs predicted to be relatively escape-resistant ($t_e^{min} > \zeta$) is shaded. **b** Location of binding residues of four HmAbs identified as relatively escape-resistant are shown as spheres on the available crystal structure (PDB ID 4MWF[43]). Heat map shows the escape times ($t_e^i$) associated with incurring mutations at antibody-binding residues similar to Fig. 3

protein. While such a simple relationship between prevalence and fitness has been reported previously for HCV NS5B[24] and many HIV proteins[25,26,28], it was not observed for the haemagglutinin protein of the influenza A virus[59]. One may ask what is the mechanistic rationale for this correspondence? Such a rationale has been proposed previously for HIV proteins, in which three key factors were identified[60]: (i) a diverse and mostly ineffective immune response due to host genetic diversity, (ii) reversion to the ancestral (fitter) sequence upon transmission to a new host, and (iii) absence of strong and effective natural or vaccine-induced herd memory responses which move the virus away from the steady state. While being a different virus, HCV is similar to HIV in many aspects and may also involve these factors. Specifically, as the majority of the sequences are sampled from chronic patients (acute HCV infections being largely asymptomatic) and E2 being a target of both neutralizing antibodies and T lymphocytes[45], it is also likely subjected to diverse and ineffective immune responses in such patients. Reversion to the consensus amino acid upon HCV transmission to a new host has also been reported[61]. Moreover, with no functioning vaccine available[1], we speculate that HCV sequences may also be in a quasi-steady state, resulting in a monotonic relationship between prevalence and fitness. The high correlation ($\bar{r} = -0.72$) observed between predicted prevalence and in vitro fitness measurements corroborates that our model can reasonably predict viral fitness (Fig. 2a), and this is on par

with that observed for HIV proteins and the HCV NS5B protein in previous studies[24–26,28].

The importance of incorporating interactions between mutations at different residues (quantified by the coupling parameters in Eq. (1)) is demonstrated by comparing our model predictions with those obtained using a simpler model based only on amino acid conservation (or single mutant probabilities), which ignores such interactions. Our tests showed that for the most direct biological validation of the model (i.e., comparison of model fitness predictions with in vitro infectivity measurements), such a conservation-only model provided a relatively lower correlation ($\bar{r} = -0.64$) (Supplementary Fig. 6; compare with the corresponding results for our model in Fig. 2a). This demonstrates the superiority of our proposed model in predicting experimental fitness. Further tests revealed that incorporating residue interactions into the model can also produce different conclusions relating to relatively escape-resistant antibodies which appear to be practically important (see Supplementary Note 1 for details).

For each E2 residue, the escape time was obtained by averaging over the time it takes to observe escape in a fixed number of transmitted/founder (T/F) sequences (see Methods). We choose this average prediction measure in order to quantify the typical time for escape from an antibody targeting a specific residue. For a given residue, a higher value of the proposed escape time metric is

indicative that antibody pressure directed toward that residue is generally more difficult to escape. We emphasize however that a high value of this metric does not preclude the existence of specific evolutionary scenarios (e.g., specific trajectories originating from specific T/F sequences) for which the escape time is fairly short, but these are expected to occur with very low probability.

Despite major advances in understanding HCV, there is still no clear consensus on whether the cellular or humoral immune response is the major determinant of spontaneous viral clearance. There are multiple reports supporting the importance of T lymphocyte-mediated immunity[46,48] while others[17,58] have demonstrated the significance of humoral immune responses in HCV clearance. Our analysis supports the possible role of humoral immunity in spontaneous viral clearance. Specifically, it suggests that E2 comprises a good proportion of exposed residues that seemingly resist mutations which, when targeted by neutralizing antibodies, may lead to clearance of infection.

A major obstacle to an effective antibody-based HCV vaccine is HVR1, the most variable and immunogenic region in E2 that elicits a nonneutralizing antibody response in the majority of infected individuals[44]. HVR1 provides shielding of important antigenic domains located in the vicinity of CD81 binding sites, which are the targets of the broadly neutralizing HmAbs analyzed in this work. These antigenic domains are believed to be disclosed only during viral entry when a conformational change takes place for the interaction of E2 with CD81[62]. Thus, a rational vaccine design should seek to elicit a repertoire of highly specific HmAbs capable of binding to the most escape-resistant regions of E2 exposed during viral entry. Recent reports show that this may be possible by using HVR1-deleted antigen as an HCV vaccine immunogen[63], by rationally designing antibodies[50] that target specifically the escape-resistant regions of E2 (e.g., those predicted in Fig. 3a), or by delivering specific HmAbs (e.g., HC33-1 and CBH-23; see Fig. 4a and Supplementary Fig. 4) using adeno-associated viral vectors[58]. In addition, it would be desirable to elicit HmAbs that target different regions of E2 to avoid competition in antibody binding[14].

While we focused primarily on genotype 1a—one of the most prevalent HCV genotypes worldwide[22]—our proposed framework can also be applied to study the E2 protein of other genotypes, provided that enough sequence data is available. Such a study can be helpful in identifying truly broadly neutralizing HmAbs (i.e., antibodies capable of neutralizing diverse HCV isolates in multiple genotypes), and thus may pave the way for the rational design of a universal prophylactic HCV vaccine.

## Methods

**Sequence data: acquisition and preprocessing**. We downloaded the amino acid MSA of HCV genotype 1a E2 from the Los Alamos National Laboratory (LANL) HCV sequence database (https://hcv.lanl.gov; accessed September 25, 2017). Specifically, searching for genotype 1, subtype a, and genomic region E2 in the search interface at https://hcv.lanl.gov/components/sequence/HCV/search/searchi.html yielded around 3400 sequences. To control sequence quality, we excluded (i) the problematic sequences (default option) which include sequences that are either very short, comprise large deletions, or are artificially synthesized in laboratories; and (ii) the sequences having more than 2% gaps in the MSA. This filtering procedure resulted in a total of $B = 3363$ sequences (accession numbers provided in Supplementary Data 2), with these sequences belonging to a total of $N = 1298$ patients. Due to this filtering procedure, the resulting sequences cover the full E2 region, i.e., there are no gaps and the majority of the sequences have no ambiguous amino acids (Supplementary Fig. 7). To control residue quality, we excluded residues which were 100% conserved (11 residues). This resulted in a total of $L = 352$ residues.

The MSA can be represented by a matrix $\mathbf{X} = \left[\mathbf{x}^{(1)}, \ldots, \mathbf{x}^{(B)}\right]^{\mathrm{T}}$ where the $k$th row is a vector of amino acids $\mathbf{x}^{(k)} = [x_1^{(k)}, \ldots, x_L^{(k)}]$ representing the $k$th sequence. Here, the amino acids at residue $i$, $x_i^{(k)}$, are identified as either consensus ($x_i^{(k)} = 0$) or the $m$th dominant (most frequent) mutant ($x_i^{(k)} = m$) for $m = 1, .., q_i$ where $q_i$ denotes the number of mutants at residue $i$.

**Fitness landscape inference model**. We infer a prevalence landscape of E2 which serves as a proxy for its fitness landscape. We choose this prevalence landscape as the "least biased" distribution that maximizes the entropy of the sequence distribution, while at the same time reproducing the single and double mutant probabilities given by

$$
\begin{aligned}
f_i(a) &= \tfrac{1}{N}\sum_{k=1}^{B} w_k \delta\left(x_i^{(k)}, a\right) \\
f_{ij}(a,b) &= \tfrac{1}{N}\sum_{k=1}^{B} w_k \delta\left(x_i^{(k)}, a\right)\delta\left(x_j^{(k)}, b\right),
\end{aligned}
\tag{2}
$$

where $f_i(a)$ denotes the frequency of amino acid $a$ at residue $i$ and $f_{ij}(a,b)$ the joint frequency of amino acids $a$ and $b$ at residues $i$ and $j$ respectively, as observed from the MSA $\mathbf{X}$. Also, $\delta$ is the Kronecker delta function

$$
\delta(a,b) = \begin{cases} 0 & \text{if } a \neq b \\ 1 & \text{if } a = b \end{cases},
\tag{3}
$$

and $w_k$ is one divided by the number of MSA sequences contributed by the patient from which sequence $k$ was extracted. We further verified that these single and double mutant probabilities are robust to the data, and are not significantly affected by the addition or removal of particular sequences in the MSA (Supplementary Fig. 8).

This least biased, or maximum entropy distribution, for a particular sequence $\mathbf{x} = [x_1, \ldots, x_L]$, assigns the probability

$$
p_{\mathbf{h},\mathbf{J}}(\mathbf{x}) = \frac{e^{-E_{\mathbf{h},\mathbf{J}}(\mathbf{x})}}{Z},
$$

with $E_{\mathbf{h},\mathbf{J}}(\mathbf{x}) = \sum_{i=1}^{L} h_i(x_i) + \sum_{i=1}^{L}\sum_{j=i+1}^{L} J_{ij}(x_i, x_j)$ and $Z = \sum_{\mathbf{x}'} e^{-E_{\mathbf{h},\mathbf{J}}(\mathbf{x}')}$. $\quad$ (4)

Here $\mathbf{h}$ denotes the set of all fields, $\mathbf{J}$ denotes the set of all couplings, $E_{\mathbf{h},\mathbf{J}}(\mathbf{x})$ is the energy of strain $\mathbf{x}$, while $Z$ is the partition function (a normalization term ensuring the distribution has total mass of unity). The field $h_i(x_i)$ and coupling $J_{ij}(x_i, x_j)$ parameters are chosen such that the single and double mutant probabilities of the model match those in the MSA, i.e.,

$$
\begin{aligned}
f_i(a) &= \sum_{\mathbf{x}} \delta(x_i, a) p_{\mathbf{h},\mathbf{J}}(\mathbf{x}) \\
f_{ij}(a,b) &= \sum_{\mathbf{x}} \delta(x_i, a)\delta\left(x_j, b\right) p_{\mathbf{h},\mathbf{J}}(\mathbf{x}).
\end{aligned}
\tag{5}
$$

This choice can be formulated as the following convex optimization problem[24]

$$
(\mathbf{h}^*, \mathbf{J}^*) = \arg\min_{\mathbf{h},\mathbf{J}} \mathrm{KL}\left(p_0 \parallel p_{\mathbf{h},\mathbf{J}}\right),
\tag{6}
$$

where

$$
p_0(\mathbf{x}) = \frac{1}{N}\sum_{k=1}^{B} w_k \delta\left(\mathbf{x}^{(k)}, \mathbf{x}\right)
$$

denotes the patient-weighted probability of observing strain $\mathbf{x}$ in the MSA, and

$$
\mathrm{KL}\left(p_0 \| p_{\mathbf{h},\mathbf{J}}\right) = \sum_{\mathbf{x}} p_0(\mathbf{x}) \ln \frac{p_0(\mathbf{x})}{p_{\mathbf{h},\mathbf{J}}(\mathbf{x})}
$$

represents the Kullback–Leibler divergence between $p_0$ and $p_{\mathbf{h},\mathbf{J}}$.

Although the optimization problem in Eq. (6) is convex and thus can be solved by gradient descent algorithms, the number of mutants and length of HCV E2 lead to computational issues. To alleviate this, we considered the inference framework introduced in ref. [28] to solve this problem accurately and efficiently. This framework comprises three steps, which are summarized in the following.

The first step in the inference framework is aimed toward limiting overfitting whilst simultaneously reducing the computational time. This is achieved by reducing the number of mutants considered per residue ($q_i$) by grouping the low-frequency mutants together. Specifically, only the top $k_i$ most-frequent mutants are considered, while the remaining $q_i - k_i$ mutants are grouped together such that the corresponding entropy with grouping achieves a certain fraction $\phi$ of the entropy without grouping. For residue $i$, this involves choosing the smallest integer $k_i$ such that

$$
S_i(k_i) \geq \phi S_i(q_i),
$$

where

$$
S_i(k_i) = -\sum_{a=0}^{k_i} f_i(a) \ln f_i(a) - \bar{f}_i \ln \bar{f}_i,
$$

and

$$
\bar{f}_i = \sum_{a=k_i+1}^{q_i} f_i(a).
$$

Now $\phi$ is chosen such that the mean of

$$
\beta_i(\phi) = \frac{\sum_{a=1}^{q_i} \left(f_i(a) - \bar{f}_i(a)\right)^2}{\sum_{a=1}^{q_i} \frac{f_i(a)(1-f_i(a))}{N}}
$$

is approximately one, where

$$\bar{f}_i(a) = \begin{cases} f_i(a) & \text{if } a < k_i + 1 \\ \bar{f}_i & \text{if } a = k_i + 1 \\ 0 & \text{if } a > k_i + 1 \end{cases}.$$

The idea is to introduce bias due to grouping (numerator), until it becomes commensurate with the fluctuations in the estimated amino acid frequencies (denominator).

Before describing the next step of the inference framework, it is convenient to define a binary matrix based on the amino acid matrix. Specifically, each amino acid at the $i$th residue is represented by $q_i$ binary digits, where $q_i = k_i + 1$ is the modified number of mutants after combining. The $j$th most-frequent amino acid is then represented by the $q_i$-bit binary representation of $2^{j-1}$, and thus the consensus sequence is the all-zero vector. We will denote $\mathbf{Y}$ as the binary matrix corresponding to the amino acid matrix after mutant combining, with $i$th row denoted by $\mathbf{y}_i$.

Note that the problem with solving Eq. (6) is that $p_{\mathbf{h},\mathbf{J}}$ contains the partition function $Z$ (see Eq. (4)), which is intractable to compute. Thus, in the second step of the inference framework, the minimum probability flow (MPF) method is used to alleviate this computational burden by replacing $p_{\mathbf{h},\mathbf{J}}$ with an alternate probability mass function (PMF). While this PMF is also parametrized by the fields and couplings, it results in a tractable optimization problem whose solution should accurately approximate the true solution. This PMF is chosen by first considering a continuous-time Markov chain whose states correspond to the $M = \prod_{i=1}^{L}(q_i + 1)$ possible sequences. The master equation describing this Markov chain is given by

$$\frac{d}{dt} p_{\mathbf{h},\mathbf{J};t}(\mathbf{y}_i) = \sum_{j=1, j\neq i}^{M} \Gamma_{ij} p_{\mathbf{h},\mathbf{J};t}(\mathbf{y}_j) - \sum_{j=1, j\neq i}^{M} \Gamma_{ji} p_{\mathbf{h},\mathbf{J};t}(\mathbf{y}_i),$$

where $p_{\mathbf{h},\mathbf{J};t}(\mathbf{y}_i)$ denotes the probability of $\mathbf{y}_i$ at time $t$, and $p_{\mathbf{h},\mathbf{J};t} = p_0$ at time $t = 0$, where $p_0$ is the empirical PMF. The solution to this master equation is given by

$$p_{\mathbf{h},\mathbf{J};t}(\mathbf{y}_i) = [\exp(t\mathbf{\Gamma})p_0]_i,$$

where $[\mathbf{a}]_i$ denotes the $i$th element of the vector $\mathbf{a}$. The matrix $\mathbf{\Gamma}$ is the $M \times M$ transition rate matrix with $(i,j)$th element $\Gamma_{ij}$ designed such

$$\lim_{t \to \infty} p_{\mathbf{h},\mathbf{J};t}(\mathbf{y}_i) = p_{\mathbf{h},\mathbf{J}}(\mathbf{y}_i),$$

with details of this matrix given in ref. [64]. The above equation implies that for any given $\mathbf{h}$ and $\mathbf{J}$, the transition rate matrix is designed to ensure that as time increases, the PMF "evolves" toward $p_{\mathbf{h},\mathbf{J}}$.

The next step is to choose a $t$ which can result in a tractable optimization problem. MPF replaces $p_{\mathbf{h},\mathbf{J}}$ in Eq. (6) by $p_{\mathbf{h},\mathbf{J};t}$ with $t$ small. As we will see, this has the advantage that the resulting optimization problem is convex and easy to solve. The resulting KL divergence between $p_0$ and $p_{\mathbf{h},\mathbf{J};t}$, expanded as a Taylor series around $t = 0$, results in the linear approximation

$$\mathrm{KL}\left(p_0 \parallel p_{\mathbf{h},\mathbf{J};t}\right) = tK_{\mathbf{h},\mathbf{J}} + o(t),$$

where

$$K_{\mathbf{h},\mathbf{J}} = \sum_{b=1}^{B} \sum_{i=1}^{L} \sum_{a=1}^{q_i} \exp\left(\frac{1}{2}\left(\left(2y_{b,(i-1)L+a} - 1\right)\sum_{j=1}^{L}\sum_{c=1}^{q_j} y_{b,(j-1)L+c}J_{ij}(a,c) - h_i(a)\right)\right),$$

with $y_{b,n}$ denoting the $(b,n)$th entry of $\mathbf{Y}$. Based on this representation, the estimate of the parameters can be found by minimizing the objective function $K_{\mathbf{h},\mathbf{J}}$. Including regularization, the estimate of the parameters is given by

$$(\mathbf{h}^{\mathrm{MPF}}, \mathbf{J}^{\mathrm{MPF}}) = \arg\min_{\mathbf{h},\mathbf{J}}\left(K_{\mathbf{h},\mathbf{J}} + \lambda_1 \sum_{i=1}^{L}\sum_{a=1}^{q_i}\sum_{j=i+1}^{L}\sum_{b=1}^{q_j}|J_{ij}(a,b)| + \lambda_2 \sum_{i=1}^{L}\sum_{a=1}^{q_i}\sum_{j=i+1}^{L}\sum_{b=1}^{q_j}J_{ij}(a,b)^2\right), \tag{7}$$

where $\lambda_1$ and $\lambda_2$ are the L1 and L2 regularization parameters, respectively, and are chosen in the next step. Note that this objective function is convex, and does not contain the intractable partition function.

For the third step of the inference framework, each field and coupling parameter set that solves Eq. (7) is used to initialize a gradient descent algorithm using MCMC simulations to approximate the gradient. Gradient descent was implemented by a modified RPROP algorithm[65], which was run for each parameter set. The couplings which were set to zero due to the L1 regularization in Eq. (7) were fixed to zero during each iteration of the gradient descent algorithm. Out of these different parameter sets, we choose the one, as in ref. [40], such that

$$\varepsilon_1 = \frac{1}{L}\sum_{i=1}^{L}\sum_{a=1}^{q_i}\frac{\left(f_i^{\mathrm{model}}(a; \lambda_1, \lambda_2) - f_i(a, \phi^*)\right)^2}{\frac{1}{N}f_i(a, \phi^*)(1 - f_i(a, \phi^*))} \approx 1, \tag{8}$$

$$\varepsilon_2 = \frac{1}{\sum_{k=1}^{L} q_k \sum_{l=k+1}^{L} q_l}\sum_{i=1}^{L}\sum_{a=1}^{q_i}\sum_{j=i+1}^{L}\sum_{b=1}^{q_j}\frac{\left(f_{ij}^{\mathrm{model}}(a,b; \lambda_1, \lambda_2) - f_{ij}(a,b,\phi^*)\right)^2}{\frac{1}{N}f_{ij}(a,b,\phi^*)\left(1 - f_{ij}(a,b,\phi^*)\right)} \approx 1, \tag{9}$$

where $f_i^{\mathrm{model}}(a; \lambda_1, \lambda_2)$ and $f_{ij}^{\mathrm{model}}(a,b; \lambda_1, \lambda_2)$ are the single and double mutant probabilities obtained from the model using regularization parameters $\lambda_1$ and $\lambda_2$ and the refined parameters from gradient descent; and $f_i(a, \phi^*)$ and $f_{ij}(a,b,\phi^*)$ are the single and double mutant probabilities after grouping with combining factor $\phi^*$. The conditions in Eqs. (8) and (9) ensure that the regularization parameters are chosen to balance overfitting and underfitting in the single and double mutant probabilities, respectively.

**Fitness verification.** To verify that the inferred prevalence landscape correlates well with the E2 fitness landscape, we compared with in vitro experimental E2 fitness measurements from literature[16,30–39]. As experimental measurements from multiple laboratories were collected, we considered the average Spearman correlation, a common approach in meta-analysis[66], which calculates the weighted average of the individual Spearman correlation $r_i$ for experiment $i$, with the weights equal to the number of fitness measurements $N_i$. This is given by

$$\bar{r} = \frac{\sum_{i=1}^{n_{\mathrm{exp}}} N_i r_i}{\sum_{i=1}^{n_{\mathrm{exp}}} N_i},$$

where $n_{\mathrm{exp}}$ is the number of experiments.

**Population genetics evolutionary model.** We consider a population genetics evolutionary model to quantify the ease of escaping immune pressure, for each residue. This is achieved by a metric, escape time, which reflects the minimum number of generations for mutations at the residue to reach a frequency > 0.5. The model incorporates the inferred fitness landscape to quantify the fitness of viral sequences. A Wright–Fisher like evolutionary model is considered (a well established population genetics model[41]) where sequences undergo mutation, selection and random sampling (genetic drift) in discrete generations. We assume a fixed sized population of $N_e = 2000$, which is in line with the effective population size of HCV for in-host evolution[67].

To calculate the number of generations for mutations to reach a frequency >0.5 for residue $i$, we first pick a sequence randomly from the MSA, with the condition that this sequence has no mutations at that residue. This sequence then forms the initial homogenous population. Sequences in this population then undergo a mutation step, where each nucleotide mutates randomly to another nucleotide with probability $\mu = 10^{-4}$, in line with known HCV mutation probabilities[5,6].

After the mutation step, each sequence undergoes selection based on the parameters of the inferred fitness model. Specifically, sequence $\mathbf{x}$ in the population will survive with probability[40,60,68]

$$f_{\mathbf{h},\mathbf{J}}(\mathbf{x}) = \frac{g_{\mathbf{h},\mathbf{J}}(\mathbf{x})}{\sum_{\mathbf{y}} g_{\mathbf{h},\mathbf{J}}(\mathbf{y})}, \text{ with } g_{\mathbf{h},\mathbf{J}}(\mathbf{x}) = \frac{e^{\beta(\bar{E} - E_{\mathbf{h},\mathbf{J}}(\mathbf{x}))}}{1 + e^{\beta(\bar{E} - E_{\mathbf{h},\mathbf{J}}(\mathbf{x}))}},$$

where $\bar{E}$ is the average energy of sequences in the population at the current generation. The function $g_{\mathbf{h},\mathbf{J}}(\mathbf{x})$ maps predicted energy (ranging from $-\infty$ to $+\infty$) to fitness by smoothly interpolating between 0, for sequences that have much lower fitness than the population average, and 1 for sequences that have much higher fitness than the population average. Note that for $g_{\mathbf{h},\mathbf{J}}(\mathbf{x})$, one can use any monotonic mapping from energy to fitness[68]. A similar form for $g_{\mathbf{h},\mathbf{J}}(\mathbf{x})$ was also used in previous Wright–Fisher like simulation models[40,60,68]. Similar to ref. [40], we estimated $\beta \sim 0.1$ based on the experimental in vitro infectivity measurements[16,30–39] and sequence energies, but we note that the qualitative results are not sensitive to the specific choice of $\beta$ (Supplementary Fig. 9a, b). To account for the effects of immune pressure on residue $i$, sequences with a wild type at that residue have their energy increased (i.e., their fitness decreased) by $b = 10$ before calculation of $f_{\mathbf{h},\mathbf{J}}(\mathbf{x})$. This value is chosen to be similar to the largest $h_j$ over all $j$, so that escape confers a selective advantage for mutants at residue $i$. Note that choosing other values of $b$ (within $\pm 10\%$ of $b = 10$) leads to similar qualitative results (Supplementary Fig. 9c, d). The reported results are robust to the value of the population size $N_e$ as well (Supplementary Fig. 9e).

As in standard Wright–Fisher models[41], individuals in the next generation are then generated through a multinomial sampling process with parameters $N_e$ and $f_{\mathbf{h},\mathbf{J}}(\mathbf{x})$. To predict the number of generations for mutations to reach a majority for residue $i$, we repeat the above evolutionary steps until the frequency of mutations at residue $i$ is over 0.5. We repeat this procedure using 25 T/F sequences from the MSA, and for each T/F sequence we perform 100 simulation runs. We then average the results to give the final escape time score $t_e^i$ for residue $i$. Note that choosing a distinct set of T/F sequences from the MSA (Supplementary Fig. 9f) or choosing T/F sequences randomly from the inferred model (equilibrium sampling via MCMC simulation; Supplementary Fig. 10) produces similar qualitative results.

**Evolutionary metrics.**

1. The minimum escape time is defined as

$$t_e^{min} = \min_i t_e^i,$$

where $i$ is either selected from the set of residues forming an antigenic domain (Fig. 3b) or the set of binding residues of an antibody (Fig. 4a).

2.  The number of residues having escape times less than or equal to a (small) threshold number of generations $\tau$ is defined as

$$n_e^\tau = \sum_i 1_{\left(t_e^i \leq \tau\right)},$$

where $i$ is selected as mentioned above, $1_{(e)}$ is the indicator function of an event $e$, and $\tau$ represents a low-escape time (around the cut-off $\zeta \sim 100$ generations) selected from the set $\{80, 100, 120, 140\}$.

**Incorporating unobserved variants to the analysis**. Our proposed method, like any data-inspired approach, is limited by the finite number of available sequences. Specifically, the proposed model cannot learn the parameters (fields $h_i$ and couplings $J_{ij}$) for any amino acid variant at a residue which is 100% conserved, and any unobserved amino acid at a residue where other variants are present in the sequence data. Nonetheless, one can still make some reasonable predictions about such unobserved variants using our model, as we describe below.

As no variation is observed at 100%-conserved residues, we excluded them during pre-processing of the data. However, this resulted in the removal of only roughly 3% of all E2 residues (11 out of 363). Moreover, given the high mutational diversity in the observed E2 sequences (i.e., with 97% of residues showing mutational variation), one reasonably expects that the lack of any observed mutations at this small number of residues is suggestive of their highly deleterious effect on viral fitness. Thus, from a vaccine perspective, these residues appear important as one would prefer to elicit an antibody response against residues where escape by incurring mutation is extremely difficult due to the associated high-fitness cost. Although we could not accurately infer the escape times associated with mutations at these 100%-conserved residues, these were incorporated in our analysis (Figs. 3 and 4) by assigning to them the largest escape time obtained for a residue ($\max_i t_e^i$) from our model. Nonetheless, even if these residues were excluded from our analysis, it does not change our findings related to antigenic domains (Fig. 3) or HmAbs (Fig. 4) as the distinguishing feature is the presence/absence of mutations associated with short-escape times.

In the comparison of model predictions with experimental fitness measurements (Fig. 2a), the infectivity of some sequences was reported with unobserved amino acid variants, and thus the energy (defined in Eq. (1)) of such sequences could not be predicted by our model. However, for including these sequences in the comparison, we assumed (similar to the above case for unobserved variants at 100%-conserved residues) that given the high genetic diversity of E2, the unobserved amino acid variant may be at least as deleterious as the least-frequent amino acid at such residues. Thus, similar to previous related works[24–28,40,60], such sequences were assigned energy values that were calculated using the same sequences, except that the involved unobserved amino acids were replaced with the least-frequent amino acids at the corresponding residues. While not being precise, this was found to be a reasonable assumption as the model predictions correlated very well with such experimental fitness measurements (see Supplementary Fig. 2).

**Relative solvent accessibility**. Each residue in the crystal structure (PDB ID: 4MWF) was assigned a solvent accessible surface area (SASA) using the PyMOL software (www.pymol.org) get_area() function, using a 1.4 solvent radius parameter. SASA values per residue were converted to relative solvent accessibility (RSA) values by normalizing by the respective SASA values per residue in a Gly-X-Gly tripeptide construct, as reported by ref. [69]. Residues with RSA > 0.2, used as a threshold in ref. [70], are considered exposed residues while the remaining residues are considered buried. In addition, the residues in HVR1 (not present in the resolved crystal structure) were also considered a part of exposed residues.

**Reporting summary**. Further information on research design is available in the Nature Research Reporting Summary linked to this article.

## Data availability

The mean escape time predicted for each residue in E2 is summarized in Supplementary Data 1. Accession numbers of genotype 1a E2 sequences used for inferring the model are listed in Supplementary Data 2. These sequences were downloaded on September 25, 2017 from the publicly available LANL HCV sequence database, https://hcv.lanl.gov. The experimental fitness (infectivity) measurements for E2 compiled from literature are included in Supplementary Data 3.

## Code availability

The software implementation of the minimum-probability-flow-based method[28], used for inferring the fitness landscape parameters, is available at https://github.com/raymondlouie/MPF-BML. Data and scripts for reproducing the results are available at https://github.com/ahmedaq/HCV-E2.

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

## Acknowledgements

We thank Arup Chakraborty, Andrew Ferguson, and John Barton for providing useful input. This research was funded by the Hong Kong Research Grant Council General Research Fund with Projects 16207915 and 16234716 (to R.H.Y.L. and M.R.M.), and a Harilela endowment (to M.R.M.).

## Author contributions

A.A.Q., R.H.Y.L. and M.R.M. designed the research, wrote the paper, and analyzed the data. R.H.Y.L. computed the fitness landscape, while A.A.Q. performed all other computations.

## Additional information

**Competing interests:** The authors declare no competing interests.

