## [Peer Review File · Nature Communications]

Reviewers' Comments:

Reviewer #1:

Remarks to the Author:

This study describes the use of sequence data to develop a fitness model for the E2 glycoprotein, which is a major target of the immune response and consequently is of interest for vaccine design efforts. While potentially interesting, it is not very clear why this model is superior to more simplistic metrics employed in previous studies to understand mAb targeting, such as level of amino acid conservation of epitope residues, particularly for the results corresponding to Figs. 2 and 3. Furthermore, as noted below, this model may overlook key rare mutations that mediate viral escape from broadly neutralizing antibodies. The above concerns and the comments below should be addressed by the authors:

1. Page 1. "HCV infection currently affects 170-180 million people worldwide". No reference is given for this number. Please provide a recent reference for this; most estimates from the past 1-2 years are lower, such as the WHO (~70 million):

<http://www.who.int/en/news-room/fact-sheets/detail/hepatitis-c>

This value may be more appropriate.

2. Page 7. Methods: " We downloaded the amino acid multiple sequence alignment (MSA) of HCV genotype 1a E2 sequences from the Los Alamos National Laboratory (LANL) HCV sequence database...". Under "HCV Sequence Alignments" in the HCV LANL database, the largest set of E2 protein alignments (dated 2008) has 627 E2 sequences, for all genotypes.

<https://hcv.lanl.gov/content/sequence/NEWALIGN/align.html>

The authors should clearly specify how they obtained over 3000 E2 sequences for gt 1a from the LANL HCV database, and how many of their sequences cover the full E2 sequence. Any redundant entries (identical in sequence) should be excluded.

3. Figure 3. The HC33.1 binding residues (spacefill in panel C) look the same as the CBH-23 binding residues. As CBH-23 is an antibody from a different antigenic domain, little or no overlap in binding residues would be expected. Plus, most HC33.1 binding residues are not resolved in the 4MWF crystal structure, as noted by the authors for Figure 2. This needs to be explained by the authors or corrected.

4. Page 5-6. In the Results section regarding the monoclonal antibody binding residue fitness, there does not seem to be any discussion of statistical significance, e.g. does CBH-23 have a statistically significant high residue fitness cost versus domain A antibodies? This should be noted if possible on the distributions in the relevant figures (Fig 2, Fig 3).

5. A major apparent shortcoming of this study is the fact that relatively rare mutations in HCV are known to occur in otherwise conserved sites due to immune pressure. These may involve some loss in viral fitness, and compensatory mutations at other sites. An example of this is the documented escape from the HCV1 monoclonal antibody in humans during a phase 2 clinical trial, through rare point mutations in antigenic domain E (Chung et al. Am. J. Transplant 2013, Babcock et al. PLoS One 2014). This is in line with epitope point mutations observed in chimpanzee models with HCV1 mAb (Morin et al, reference 15 in the manuscript). The two human studies noted above should be referenced, and this possible limitation in the premise should be addressed, in the Discussion at very least.

Reviewer #2:

Remarks to the Author:

The broad aims of the paper are to empirically determine the fitness landscape of the epitope regions of the E2 envelope protein of HCV and use this landscape to infer sites which have large fitness costs as good candidates for synthetic vaccines.

* The authors use a maximum entropy model inspired from the statistical physics to empirically determine an effective energy function by finding optimum parameters of this function so as to match single and double mutant probabilities from a multiple sequence alignments.

* The rationale behind this approach is sound, and asks what is the distribution of maximum uncertainty given the constraints from the data (here the single & double mutant probabilities) and assuming nothing else is known.

* The energy function they use is an extension of the Ising model called a Potts model, that allows more than two states (here amino acids) at each site - it has a single site contribution and pair-wise contribution between sites.

* Determining the optimum set of parameters is computationally challenging, for which they use technique called "minimum probability flow", which is used to minimise the relative entropy between empirical and theoretical mutant probabilities - they say this min probability flow method allows this to be done efficiently and accurately - however, no details are given and instead a reference to a previous paper of the authors on HIV is given.

* They define a measure of fitness cost DE_i for each residue which is the *average* fitness change when the amino acid is changed to anything other than the consensus, where the average is taken over all possible backgrounds (Eqn 10 in Methods)

* They validate their empirical energy function and fitness cost as a good proxy for the fitness landscape using various measures in Fig 1.

* They then use their computational technique to assess the fitness cost of various regions of which have been determined to have been bound by HmAbs, identifying regions which using the fitness cost measure DE_i predicts that there are no low fitness cost mutations.

My main criticisms are

* the procedure cannot learn anything about the evolution of the system, more than can be obtained given the finite sample of sequences - rare variants in the system at a frequency much less than the inverse of the sample size cannot be assigned h_i or J_{ij} values

* In principle, this gives $20 \times 350 = 7000$ single site fitness coefficients and $\sim 20^2 \times 350^2 / 2 \sim 24$ million pair-wise coefficients to be determined from only 3000 sequences - this clearly is impossible. It is not clear what their assumptions are in this regard. In any case they can only determine coefficients in the model for amino acids that are actually observed in this sample - it is possible that they assume anything below their detection threshold is not important - some discussion of this assumption would be good along with some idea of how much variation they are missing, giving known values of $\mu \times N$.

* A correlation of 0.74 is good, but given that even triple mutant probabilities are reproduced (which suggests triple mutant probabilities are products of combinations of double and single mutant effects) - so why not a better correlation - what is missing in the model?

* DE_i is calculated by summing over all other sequences possible except at focal site i (background sequences) - there must be specific background sequences that contribute most to the average DE_i - are they predominantly single effect, or double/triple or higher order mutants?

* DE_i is also discussed in the absence of what is the relevant evolutionary scale of fitness or of their energy measure/fitness cost, which is $1/N$, where N is the effective population size

* This leads to the key problem I have with this computational technique, is that this analysis could mask particular combinations of background sequences that are of low fitness cost, despite the overall average at that site being of high fitness cost. Hence the mean itself is potentially meaningless without some idea of the distribution of fitness effects at each site/residue. What is important is something like the minimum fitness cost at each site when examined over all background sequences - even if a single mutational path exists of low fitness cost then it is just a question of how long it takes for that path to arise. In practice, if it requires many changes at other sites, then it may not be likely to arise on relevant time scales - however, if it requires a double or triple mutant across other sites then this is could be arise quickly given a large mutational input ($N \cdot \mu \gg 1$)

* So in the figures in Fig.2 which seems to show box plots over regions of the protein, these give an indication of the variance of the mean fitness costs over a number of residues and so also mask potential low fitness cost mutants.

* To address this there would need to be some analytical or simulation analysis, using their empirically determined fitness function that confirms or not that regions identified as being "free" of low fitness mutations indeed do not develop escape mutations in concert with other changes in the protein. There are various options for the simulations, but they would need to allow there to be polymorphisms in the population - so for example individual based simulations. The tricky question in the simulations is what effective population size should be used. I do not know of any studies of N in HCV, but in HIV it has consistently been measured to be of order $\sim 10^4$ or 10^5 , thought to be reflective of population structure and maybe more simply that only infected and actively replicating cells produce genetic variation, and so the effective population size is of order the number of infected cells.

More minor comments:

* I could not find discussion of the lower right figure of Fig.1a in the main text

* Their minimum probability flow method should be outlined in the methods and or SI, as it is a new and relatively non-standard optimisation technique.

RESPONSE TO REVIEWERS' COMMENTS

Identifying immunologically-vulnerable regions of the HCV E2 glycoprotein and broadly neutralizing antibodies that target them (NCOMMS-18-11143)

We are gratified by the positive comments and recommendations of both reviewers. We especially thank the reviewers for the time they devoted to provide detailed and thoughtful reviews, which have helped to improve our paper. Below, we address each of the reviewers' comments and note the changes made to the manuscript in response (in blue).

REVIEWER 1

General comments:

1. *This study describes the use of sequence data to develop a fitness model for the E2 glycoprotein, which is a major target of the immune response and consequently is of interest for vaccine design efforts. While potentially interesting, it is not very clear why this model is superior to more simplistic metrics employed in previous studies to understand mAb targeting, such as level of amino acid conservation of epitope residues, particularly for the results corresponding to Figs. 2 and 3.*

Response:

We agree that it is important to compare our predictions with those of a simpler model based only on amino acid conservation (or single mutant probabilities) which ignores interactions between residue mutations. The most meaningful comparison is with respect to a conservation-based maximum entropy model parametrized only by the "fields" $h_i(a)$. These are given by

$$h_i(a) = \log \frac{f_i(a)}{1 - f_i(a)},$$

where $f_i(a)$ is the frequency of observing mutant a at residue i . While we did not overly emphasize this point, in the Results section of the original manuscript we had briefly mentioned a comparison with this model, at least in terms of fitness predictions. Specifically, our tests indicated that the correlation between the fitness predictions using this model and in vitro infectivity measurements ($\bar{r} = -0.61$; Supplementary Fig. S3a, reproduced below for convenience) was markedly lower than that obtained using our proposed model ($\bar{r} = -0.74$; see Fig. 2a in the revised manuscript¹). This demonstrates the superiority of the proposed model in predicting experimental fitness. It also attests to the importance of incorporating interactions between mutations at different residues (quantified by the coupling parameters in equation 1). This is consistent with experimental observations confirming that epistatic interactions play a major role in HCV evolution and immune escape (Campo et al. 2008; Oniangue-Ndza et al. 2011; Parera & Martinez 2014).

We chose to initially compare the models in terms of their prediction accuracy against fitness experiments only, as this is the most direct biological validation. Nonetheless, we appreciate the reviewer's point, and agree that an important question remains as to whether the improved fitness predictions of our model over a simple conservation-only model translate to non-trivial differences in the subsequent analysis and conclusions; namely, for the classification of antigenic domains (Fig. 3) and antibodies (Fig. 4). We have now investigated this in detail.

¹ Note that all references to the main-text and supplementary figures are according to their numbering in the revised manuscript. For example, Figs. 2 and 3 in the original submitted manuscript are referred to as Figs. 3 and 4 in the current revised version.

Figure S3 | Experimental fitness vs predicted energy for the conservation-only model (a) Normalized experimental fitness vs predicted energy (similar to Fig. 2a). The correlation \bar{r} is the weighted Spearman correlation coefficient (see Methods).

We first computed the analogous result to Fig. 3 (fitness costs associated with mutating residues in different antigenic regions of E2), using the conservation-only model. These results are shown in Fig. R1, where they are contrasted against the predictions of our proposed model. Some important differences are evident.

Most notably, the absence of low-fitness-cost (easy escape) mutations in our model predictions for domains C and E clearly differentiates them from the remaining domains (Fig. R1, left panel). Such an apparent difference is not evident from the conservation-only model (Fig. R1, right panel). This prediction of our model for domains C and E to be robust to escape is consistent with their known important role in HCV-CD81 binding (Pierce et al. 2016). Moreover, our model predicted multiple low-fitness-cost mutations in domain A—known to be highly flexible and non-neutralizing (Keck et al. 2005)—which makes it clearly distinct from the robust domains C and E (Fig. R1, left panel); such distinction is not so evident from the conservation-only model (Fig. R1, right panel), particularly in regards to domains A and E.

Another key observation is that the two models produced quite distinct predictions for domain D. While the conservation-only model (Fig. R1, right panel) predicted the fitness costs of mutations in domain D to be high (even slightly higher than those in domain E); our model (Fig. R1, left panel) predicted domain D to comprise low fitness cost mutations, implying the possibility of viral escape from antibodies targeting this domain. This prediction of our model is consistent with a recent experimental study (Bailey et al. 2015) reporting residue 442 in domain D as one of the common mutations in E2 that confer resistance to multiple neutralizing antibodies. Our model predicted the mutations at this residue to incur low fitness cost ($\Delta E_i \sim 0$). Note that the identification of antigenic domains that are devoid of any low-fitness-cost mutation is particularly useful for vaccine development as one can rationally design potentially escape-resistant antibodies (Sormanni et al. 2015) that specifically bind to such E2 domains. Generally, these differences demonstrate that predictions from our model are comparatively more in line with experimental reports than those using a simple conservation-only model.

Figure R1 | Analysis of fitness costs associated with mutating the residues in E2 regions that are targeted by antibodies.

Results using the proposed model (*left panel*) and the conservation-only model (*right panel*). Antigenic domains C and E (shaded in the left panel) are predicted by the proposed model to be the only domains devoid of low-fitness-cost mutations.

We next computed the analogous result to Fig. 4 (fitness costs of mutating the binding residues of HmAbs), using the conservation-only model. These results are shown in Fig. R2, where they are contrasted against the predictions of our proposed model. Once again, important differences are revealed.

A first key difference is the clear distinction of the two groups of HmAbs—(i) with low-fitness-cost-mutations ($\Delta E_i \sim 0$) and (ii) with no low-fitness-cost mutations—apparent from our model predictions (Fig. R2a, left panel). Barring HmAb HC33-1, the same cannot be said for the predictions using the conservation-only model in general (Fig. R2a, right panel).

A second key observation is that the predictions of the two models are quite distinct for domain D HmAb HC84-20 (Fig. R2a). Specifically, at the stricter and more commonly used threshold for defining binding-residues ($RB \leq 20\%$), while the conservation-only model predicted HC84-20 to comprise multiple binding residues with low-fitness-cost mutations (non-conserved), our model identified it as a robust HmAb devoid of low-fitness-cost mutations (Fig. R2a, top panel). This distinction in our model prediction suggests that the mutations at the non-conserved binding residues of HC84-20 still bear high-fitness-cost due to the additional constraints imposed by their interaction with other protein residues (that are taken into account exclusively in our model). Note that HmAb HC84-20 is different from the other domain D HmAbs (HC84-24 and HC84-26) as residue 442—a common binding residue of these HmAbs—is not a binding residue of HC84-20 at $RB \leq 20\%$ (Pierce et al. 2016). Mutations at the residue 442 are known to be associated with escape from domain D HmAbs (Bailey et al. 2015) and our model correctly associated it with low-fitness-cost mutation. This is evident from our model predictions at $RB \leq 40\%$ (Fig. R2a, bottom left panel) where residue 442 is a binding residue of all domain D HmAbs (including HC84-20), and at $RB \leq 20\%$ (Fig. R2a, top left panel) where it is a binding residue only of HmAbs HC84-24 and HC84-26. Thus, the absence of residue 442 from the binding residues of HmAb HC84-20 at $RB \leq 20\%$ suggests that this domain D HmAb may be a promising candidate for vaccine design, which also corroborates the predictions of a recent report (Cowton et al. 2018).

A third key difference is that contrary to the conservation-only model's prediction, our model predicted no low-fitness-cost mutations in the binding residues of the HmAbs AR3A-AR3C (Fig. R2b), suggesting the potential difficulty for the virus in escaping these antibodies. This is consistent with the reported potency of these antibodies in preventing as well as clearing chronic HCV infection in humanized mice (de Jong et al. 2014). Moreover, no escape mutations have been reported in studies related to these antibodies as of yet (de Jong et al. 2014; Velázquez-Moctezuma et al. 2017).

Figure R2 | Analysis of fitness costs associated with mutating the binding residues of known HmAbs.

Results using the proposed model (*left panel*) and the conservation-only model (*right panel*). (a) Fitness cost (in terms of ΔE_i) associated with mutating the binding residues of 16 HmAbs, determined using global alanine scanning (Pierce et al. 2016). The binding residues of the HmAbs are defined as the mutations that resulted in reduction of binding with respect to the wild-type (RB) to $RB \leq 20\%$ (*top panel*) and $RB \leq 40\%$ (*bottom panel*). HmAbs are colored according to the antigenic domains using the scheme in Fig. 3B. The background of the HmAbs predicted by our proposed model to comprise residues with no low-fitness-cost mutations is shaded grey (*left panels*). (b) Fitness cost associated with mutating the binding residues of HmAbs, where binding residues were determined using selective alanine scanning mutagenesis (*Supplementary Table S3*). All HmAbs predicted to have no low-fitness-cost mutations in the binding residues are grouped together (in no specific order) on the left (with grey background) and the remaining ones (with at least one residue with low-fitness-cost mutations) are grouped together on the right (with white background).

Collectively, these results reveal that incorporating residue interactions into our model is not only important in making fitness predictions, but can also produce different conclusions relating to antibody escape which appear to be practically important. We thank the reviewer for raising this important point, and for providing the impetus for conducting a more thorough comparative study

with respect to simple conservation-based models. We have included a summary of these results in the Discussion section, while the detailed comparison has been incorporated in the Supplementary Text S1.

2. *Furthermore, as noted below, this model may overlook key rare mutations that mediate viral escape from broadly neutralizing antibodies.*

Response:

Please refer to below, where we address this point when responding to the last specific comment (comment 5).

Specific comments:

1. *Page 1. "HCV infection currently affects 170-180 million people worldwide". No reference is given for this number. Please provide a recent reference for this; most estimates from the past 1-2 years are lower, such as the WHO (~70 million): <http://www.who.int/en/news-room/fact-sheets/detail/hepatitis-c>. This value may be more appropriate.*

Response:

We thank the reviewer for pointing this out. The reported statistic was taken from an older WHO bulletin <http://www.who.int/bulletin/volumes/90/7/11-097147/en/> and we agree that the more recent estimates are lower. We have updated this statement and provided proper reference in the revised manuscript.

2. *Page 7. Methods: "We downloaded the amino acid multiple sequence alignment (MSA) of HCV genotype 1a E2 sequences from the Los Alamos National Laboratory (LANL) HCV sequence database...". Under "HCV Sequence Alignments" in the HCV LANL database, the largest set of E2 protein alignments (dated 2008) has 627 E2 sequences, for all genotypes. <https://hcv.lanl.gov/content/sequence/NEWALIGN/align.html>. The authors should clearly specify how they obtained over 3000 E2 sequences for gt 1a from the LANL HCV database, and how many of their sequences cover the full E2 sequence. Any redundant entries (identical in sequence) should be excluded.*

Response:

We understand that we had not provided sufficient details regarding the procedure for downloading the HCV E2 genotype 1a sequence data used for our analysis. This data was downloaded from the search interface available at the following link on HCV LANL: <https://hcv.lanl.gov/components/sequence/HCV/search/searchi.html>. We had also checked the link mentioned by the reviewer (<https://hcv.lanl.gov/content/sequence/NEWALIGN/align.html>), and as stated, it returned far less sequences. The main reason is that this specific page of the database has not been updated since 2008, whereas the page which we used has been progressively updated.

Searching for genotype "1", subtype "a", and genomic region "E2" yielded around 3400 sequences. As mentioned in the manuscript, to ensure sequence quality we excluded (i) the problematic sequences (default option) which include sequences that are either very short, comprise large deletions, or are artificially synthesized in laboratories; and (ii) the sequences having more than 2% gaps in the MSA. This filtering procedure resulted in a total of 3363 sequences, and these were the specific sequences analyzed in our work. Moreover, we have provided the accession numbers of these sequences, which can be used to access them from the GenBank (<https://www.ncbi.nlm.nih.gov/genbank/>). Due to the above filtering procedure, the resulting sequences cover the full E2 region; there are no gaps in these sequences and the majority of them have no ambiguous amino acids (Fig. R3).

We additionally point out that we do not exclude identical sequences from our data. This is because the presence of an identical sequence in different patients appears informative in terms of fitness. Nevertheless, many sequences (identical or not) taken from the same patient may bias the inferred model and may affect the ability of the model to predict intrinsic fitness. This is a potential problem for E2 based on the available sequence data: the filtered 3363 sequences belong to 1298 patients only. To reduce patient-specific biases, in line with previous related studies (Barton et al. 2016; Louie et al. 2018) we introduced appropriate weights (w_k) in our inference procedure while calculating the single and double mutant probabilities observed in the MSA (equation 3). Specifically for sequence k , w_k is set to one divided by the number of MSA sequences contributed by the patient from which that sequence was extracted; thus, penalizing the contribution of mutants observed in the sequences that are extracted from the same patient.

To clarify the above points, we have revised the text in the Methods section. Fig. R3 has also been included in the supplement and is referred to as Supplementary Fig. S7.

Figure R3 | The fraction (per sequence) of gaps and ambiguous amino acids in the processed E2 sequence data.

In this box plot, the bold horizontal line indicates the median, the edges of the box represent the first and third quartiles, whiskers extend to span a 1.5 inter-quartile range from the edges (both box edges and whiskers are not visible due to the distribution being too skewed around the median), and the spheres represent the outliers (values beyond the whiskers' range).

3. *Figure 3. The HC33.1 binding residues (spacefill in panel C) look the same as the CBH-23 binding residues. As CBH-23 is an antibody from a different antigenic domain, little or no overlap in binding residues would be expected. Plus, most HC33.1 binding residues are not resolved in the 4MWF crystal structure, as noted by the authors for Figure 2. This needs to be explained by the authors or corrected.*

Response:

We thank the reviewer for pointing this out. This was a typographical oversight during the original submission. The binding residues of HmAb HC33.1 are indeed different from those of HmAb CBH-23. We have rectified this error and incorporated the correct figure into the revised manuscript. This is shown below for convenience.

Revised Figure 4c: (previously Figure 3c)

Figure 4 | Analysis of fitness costs associated with mutating the binding residues of known HmAbs identifies escape-resistant HmAbs. [...] (c) Location of binding residues (determined using global (top panel) and selective (bottom panel) alanine scanning mutagenesis) of selected HmAbs identified as escape-resistant are shown as spheres on the available crystal structure (PDB ID 4MWF). Heat map shows the fitness cost of mutations.

4. Page 5-6. In the Results section regarding the monoclonal antibody binding residue fitness, there does not seem to be any discussion of statistical significance, e.g. does CBH-23 have a statistically significant high residue fitness cost versus domain A antibodies? This should be noted if possible on the distributions in the relevant figures (Fig 2, Fig 3).

Response:

Our model predictions showed that the salient feature which clearly distinguished the known non-neutralizing domain A HmAbs (Keck et al. 2005) from other neutralizing HmAbs was the abundance of binding residues with low-fitness-cost mutations. In the paper, we focused on using this feature for the classification of known antigenic domains (Fig. 3) and HmAbs (Fig. 4) into two categories: escape-resistant and escape-susceptible. However, looking at our model predictions in Fig. 4, the distributions of the fitness costs associated with binding-residue mutations of all HmAbs appear broadly similar, with the key differences corresponding to the low-fitness-cost mutations appearing only in the tail of the distribution (i.e., mainly as outliers). Due to the broad similarity of the distributions, if one applies a standard statistical test to identify if the fitness cost associated with a non-domain A HmAb is higher than that of domain A HmAbs, the results do not appear statistically significant ($P > 0.05$ for all such pair-wise comparisons of HmAbs; Mann-Whitney test). The problem here is that such a statistical test compares the similarity of entire distributions, and is not suitable for showing significance of the distinguishing feature that we are interested in; i.e., an abundance of binding residues with low-fitness-cost mutations.

As an alternative, we investigated whether we could statistically quantify the importance of these low-fitness-cost mutations using outlier-detection-related statistical tests, like the Tietjen-Moore test (Tietjen & Moore 1972) and the generalized extreme Studentized deviate (ESD) test (Rosner 1983). These, however, were also not found to be suitable, since they are parametric tests based on assumptions that are inappropriate for our case (Gaussianity of data, a priori information about the outliers, etc.).

Nonetheless, despite lacking a suitable statistical measure for ascribing a mathematical significance value to the low-fitness-cost mutations, we build confidence in the ability of our model to predict low- and high-fitness-cost mutations through comparison with multiple biological data (Fig. 2). Specifically, (i) our model predicted known escape-associated residues from antibodies to be associated with low-fitness-cost mutations as compared to the remaining residues, which were largely predicted to be associated with high-fitness cost mutations (Fig. 2b). Similarly, (ii) it correctly predicted the surface and buried residues to comprise generally low-fitness-cost and high-fitness-cost mutations, respectively (Fig. 2d). (iii) It correctly associated the mutations in the T cell epitopes known to be targeted by spontaneous-clearance-associated HLA alleles (McKiernan et al. 2004; Singh et al. 2007; Kim et al. 2011) with high-fitness-cost as compared to those targeted by other alleles (Fig 2e; reproduced here in Fig. R4a for convenience). Importantly, for this biological validation, no low-fitness-cost mutation was predicted to be present in the set of spontaneous-clearance-associated epitopes. We explored this further to obtain more precise information about T cell escape in individual epitopes, which gave even stronger (and quite direct) validation of the ability of our model to correctly distinguish those residues which have low-fitness-cost mutations from the others. Specifically, of the five epitopes predicted by our model to be devoid of low-fitness-cost mutations ($\Delta E_i \sim 0$), four epitopes were restricted by spontaneous-clearance-associated HLA alleles (Prediction accuracy = 0.93; Fig. R5c). (We point out that such an accurate and robust classification was not revealed with the conservation-only model (Fig. R5). Thus, adding to our first response, this result provides further confidence in our model predictions as compared to the conservation-only model.)

The above-mentioned results have now been incorporated in the Supplementary Text S1.

Figure R4 | Prediction of epitopes associated with spontaneous clearance using our model. (a) Comparison of the distribution of fitness costs associated with mutations in the epitopes targeted by spontaneous-clearance-associated HLA alleles and those targeted by other alleles (b) Distribution of the fitness costs associated with mutations in the 16 individual E2-restricted epitopes. The epitopes targeted by spontaneous-clearance-associated HLA alleles are colored red while the remaining epitopes are colored blue. The background of the epitopes predicted by our model to be potentially associated with spontaneous clearance is shaded.

Figure R5 | Prediction of epitopes associated with spontaneous clearance using the conservation-only model.

(a) Comparison of the distribution of fitness costs associated with mutations in the epitopes targeted by spontaneous-clearance-associated HLA alleles and those targeted by other alleles (b) Distribution of the fitness costs associated with mutations in the 16 individual E2-restricted epitopes. The epitopes targeted by spontaneous-clearance-associated HLA alleles are colored red while the remaining epitopes are colored blue. (c) Comparison of the accuracy of predicting epitopes associated with spontaneous clearance using our proposed model and the conservation-only model. Accuracy is defined as $\frac{1}{L} [\sum TP + \sum TN]$, where L is the total number of epitopes, and TP and TN is the number of true positives (correctly predicted spontaneous-clearance-associated epitopes) and the number of true negatives (correctly predicted epitopes that are not associated with spontaneous clearance), predicted at threshold x (used to define the low-fitness-cost mutations, i.e., $\Delta E_i \leq x$), respectively.

5. A major apparent shortcoming of this study is the fact that relatively rare mutations in HCV are known to occur in otherwise conserved sites due to immune pressure. These may involve some loss in viral fitness, and compensatory mutations at other sites. An example of this is the documented escape from the HCV1 monoclonal antibody in humans during a phase 2 clinical trial, through rare point mutations in antigenic domain E (Chung et al. *Am. J. Transplant* 2013, Babcock et al. *PLoS One* 2014). This is in line with epitope point mutations observed in chimpanzee models with HCV1 mAb (Morin et al, reference 15 in the manuscript). The two human studies noted above should be referenced, and this possible limitation in the premise should be addressed, in the Discussion at very least.

Response:

It is true that if no variation is observed at a residue, the fitness cost of mutations at that residue cannot be reliably quantified with our model. (The same is in fact true for any other data-inspired model.) We excluded any such 100%-conserved residues during pre-processing of the data (indicated in Methods); however, this resulted in the removal of only roughly 3% of all E2 residues (11 out of 363). Moreover, given the high mutational diversity in the observed E2 sequences (i.e., with 97% of residues showing mutational variation), one reasonably expects that the lack of any observed mutations at this small number of residues is suggestive of their highly deleterious effect on viral fitness. Thus, from a vaccine perspective, these residues appear important as one would prefer to elicit an antibody response against residues where any mutation incurs a high fitness cost

to the virus. Although we could not precisely infer the fitness cost associated with these 100%-conserved residues using our model, these were still ultimately incorporated into our analysis (Figs. 3 and 4) by assigning to them the highest (worst) fitness-cost of mutation obtained for a residue ($\max_i \Delta E_i$) from our model.

We also point out that even if we had completely excluded these residues from the analysis (rather than incorporating them as described above), it does not change our main conclusions related to antigenic domains (Fig. 3) or HmAbs (Fig. 4), as the distinguishing feature is the presence/absence of *low*-fitness-cost mutations.

We thank the reviewer for mentioning this important point related to rare mutations observed in HCV (which has been mentioned by reviewer 2 as well). Although we were accounting for these rare mutations in our analysis in the manner described above, we recognize that the related discussion was missing. We have added this detail in the Methods section of the revised manuscript.

Regarding the specific escape example mentioned by the reviewer, we note that our means of handling 100%-conserved residues just described was not relevant for this example. To elaborate, the residues 415 and 417 involved in escape from HmAb HCV1 (binding residues 412-423), as reported in (Morin et al. 2012) and (Chung et al. 2013; Babcock et al. 2014), while being highly conserved (residues 415 and 417 were 98.9% and 99.2% conserved respectively), the small variation at these residues meant that they were not excluded during the initial screening of 100%-conserved residues. Our model can therefore associate fitness costs with mutations at these residues. To address the reviewer's comment, we specifically investigated the fitness cost predicted by our model for (i) the rare HCV1 epitope escape mutations (at residues 415 and 417) and the accompanying non-epitope mutations reported (ii) in (Morin et al. 2012) and (iii) in (Babcock et al. 2014).

First, we investigated the fitness cost associated with mutating all the binding residues of HmAb HCV1 in isolation. Our model predicted the two escape mutations (at residues 415 and 417) to be among the five easiest escape mutations possible in the binding residues of HCV1 (Fig. R6). The relatively high ΔE_i values predicted by our model for the HCV1 escape mutations ($\Delta E_i \sim 6$) suggest that these may not be as easy to make as those observed in other antibodies (with associated $\Delta E_i \sim 0$). Indeed, if these are rare mutations, as indicated by the reviewer, then it is highly possible that these mutations are deleterious for the virus. It is thus possible that under immune pressure from HmAb HCV1, the virus may eventually take this escape pathway to survive at a relatively lower fitness. Our prediction of antigenic domain E to be escape-resistant implies that evading the antibodies binding to this domain may be more difficult than those binding to other escape-susceptible domains. Interestingly, the binding residues of another domain E HmAb HC33-1 does not involve these residues (415 and 417) on which escape mutation was reported in the case of HmAb HCV1 (Fig. R7), and our analysis indicates that HC33-1 may be comparatively more robust than HCV1. In addition, these reports (Morin et al. 2012; Chung et al. 2013; Babcock et al. 2014) studied the effect of specifically introducing only HmAb HCV1 in humans and chimpanzees. However, for an effective vaccine, inducing multiple identified escape-resistant HmAbs (including HCV1) simultaneously may help in minimizing the possibility of viral escape by mounting an antibody response directed towards different E2 regions.

We have included this analysis of the observed HCV1 escape mutations in the Discussion section of the paper, as suggested by the reviewer.

Figure R6 | Fitness cost (in terms of ΔE_i) associated with mutations on binding residues of HmAb HCV1.

Figure R7 | Comparison of fitness costs (in terms of ΔE_i) associated with mutating the binding residues of two domain E HmAbs, namely HCV1 and HC33-1.

Second, we investigated the ability of our model to predict the non-epitope compensatory mutation (Q444R) reported to be associated with the escape mutation (N417S) that confers resistance to HmAb HCV1 in a chimpanzee model (Morin et al. 2012). The compensatory effect of Q444R for N417S was confirmed by engineering these mutations in the H77 virus E2 glycoprotein in vitro. To compare with the results of this experiment, we analyzed the change in energy observed with our model by making all possible mutations in the H77 strain carrying the N417S mutant (H77_{N417S}). Mutations yielding a negative change in energy imply an increase in fitness, whereas those yielding a positive energy change would imply a fitness decrease. Interestingly, our model predicted four mutations at residue 444 (Q444H, Q444V, Q444Y, and Q444R) to be associated with the **most negative change in energy** among all mutations in the H77_{N417S} strain (Fig. R8).

This is an important example further demonstrating the ability of our model to predict possible non-epitope compensatory mutations, and we thank the reviewer for guiding us to explore this aspect in detail. We have included this result as an additional model validation (Fig. 2c) and have cited the related studies in the revised manuscript.

Figure R8 | Histogram of the change in energy observed by all possible mutations “X” in the H77 strain carrying the N417S mutant (H77_{N417S}).

Mutations with the most negative change in energy (i.e., the highest compensatory effect) are labeled on the plots. The Q444R mutation reported to be compensating for N417S (Morin et al. 2012) is shown in bold.

In addition to mutations on residues 415 and 417 (Chung et al. 2013), in the alternative study (Babcock et al. 2014) indicated by the reviewer, mutations on multiple non-epitope residues were observed during a phase 2 clinical trial in humans administered with HmAb HCV1. However, in this case, the possible compensatory role of these mutations is not clear. This is because in vitro experiments—similar to those conducted in (Morin et al. 2012)—were not performed to confirm compensatory fitness effects. (Interestingly the Q444R mutation, reported to be compensatory for the HCV1 escape mutation N417S (Morin et al. 2012), was not reported in this study.) Moreover, information regarding the actual background strains upon which these compensatory mutations appeared were not reported, and consequently, an analysis similar to Fig. R8 could not be performed. (We note that the results of our model using H77_{N417S} or H77_{N415K} as a background strain (similar to Fig. R8) showed no compensatory effect of these mutations.) Nonetheless, computing the average fitness cost of mutations at these reported residues in an ensemble of potentially viable viral strains (similar to Fig. 2b) indicated that they appear broadly associated with low-fitness-cost mutations, at least as compared to the remaining protein residues (Fig. R9). This gives a (albeit rather weak) suggestion that mutations at these residues may possibly assist escape through compensation. An alternative possibility is that they may simply be random low-fitness-cost mutations hitchhiking on fit backgrounds. In general, comprehensive in vitro experiments, similar to (Morin et al. 2012), are still required to unambiguously confirm the role of these non-epitope mutations in conferring escape from HCV1.

Figure R9 | Comparison of the distribution of fitness costs (in terms of ΔE_i) associated with mutations on non-epitope residues, observed during a phase 2 clinical trial in humans administered with HmAb HCV1 (Babcock et al. 2014), and that of the remaining protein residues.

REVIEWER 2

Summary:

The broad aims of the paper are to empirically determine the fitness landscape of the epitope regions of the E2 envelope protein of HCV and use this landscape to infer sites which have large fitness costs as good candidates for synthetic vaccines. The authors use a maximum entropy model inspired from the statistical physics to empirically determine an effective energy function by finding optimum parameters of this function so as to match single and double mutant probabilities from a multiple sequence alignments. The rationale behind this approach is sound, and asks what is the distribution of maximum uncertainty given the constraints from the data (here the single & double mutant probabilities) and assuming nothing else is known. The energy function they use is an extension of the Ising model called a Potts model, that allows more than two states (here amino acids) at each site - it has a single site contribution and pair-wise contribution between sites. Determining the optimum set of parameters is computationally challenging, for which they use technique called "minimum probability flow", which is used to minimise the relative entropy between empirical and theoretical mutant probabilities - they say this min probability flow method allows this to be done efficiently and accurately - however, no details are given and instead a reference to a previous paper of the authors on HIV is given. They define a measure of fitness cost DE_i for each residue which is the *average* fitness change when the amino acid is changed to anything other than the consensus, where the average is taken over all possible backgrounds (Eqn 10 in Methods). They validate their empirical energy function and fitness cost as a good proxy for the fitness landscape using various measures in Fig 1. They then use their computational technique to assess the fitness cost of various regions of which have been determined to have been bound by HmAbs, identifying regions which using the fitness cost measure DE_i predicts that there are no low fitness cost mutations.

Specific comments:

1. The procedure cannot learn anything about the evolution of the system, more than can be obtained given the finite sample of sequences - rare variants in the system at a frequency much less than the inverse of the sample size cannot be assigned h_i or J_{ij} values.

Response:

It is true that our proposed method, like any data-inspired approach, is limited by the finite number of available sequences. Specifically, the proposed model cannot learn the parameters (fields h_i and couplings J_{ij}) for: (i) any amino acid variant at a residue which is 100% conserved, and (ii) any unobserved amino acid at a residue where other variants are present in the sequence data. Nonetheless, one can still make some reasonable predictions about such unobserved variants using our model, as we describe below.

As no variation is observed at 100%-conserved residues, we excluded them during pre-processing of the data (indicated in Methods). However, this resulted in the removal of only roughly 3% of all E2 residues (11 out of 363). Moreover, given the high mutational diversity in the observed E2 sequences (i.e., with 97% of residues showing mutational variation), one reasonably expects that the lack of any observed mutations at this small number of residues is suggestive of their highly deleterious effect on viral fitness. Thus, from a vaccine perspective, these residues appear important as one would prefer to elicit an antibody response against residues where any mutation incurs a high fitness cost to the virus. Although we could not accurately quantify the fitness cost associated with mutations at these 100%-conserved residues, these were incorporated in our analysis² (Figs. 3 and 4) by assigning to them the highest (worst) fitness-cost of mutation obtained for a residue ($\max_i \Delta E_i$) from our model. Nonetheless, even if these residues were excluded from our analysis, it does not change our findings related to antigenic domains (Fig. 3) or HmAbs (Fig. 4) as the distinguishing feature is the presence/absence of easy escape mutations.

² Note that all references to the main-text and supplementary figures are according to their numbering in the revised manuscript. For example, Figs. 2 and 3 in the original submitted manuscript are referred to as Figs. 3 and 4 in the current revised version.

In the comparison of model predictions with experimental fitness measurements (Fig. 2a), the infectivity of some sequences was reported with unobserved amino acid variants, and thus the energy (defined in equation 1) of such sequences could not be predicted by our model. However, for including these sequences in the comparison, we assumed (similar to the above case for unobserved variants at 100%-conserved residues) that given the high genetic diversity of E2, the unobserved amino acid variant may be at least as deleterious as the least-frequent amino acid at such residues. Thus, similar to previous related works (Ferguson et al. 2013; Shekhar et al. 2013; Mann et al. 2014; Barton et al. 2015; Hart & Ferguson 2015; Barton et al. 2016; Flynn et al. 2017; Louie et al. 2018), such sequences were assigned energy values that were calculated using the same sequences, except that the involved unobserved amino acid was replaced with the least-frequent amino acid at the corresponding residue. While not being precise, this was found to be a reasonable assumption as the model predictions correlated very well with such experimental fitness measurements (Fig. R10). For example, infectivity measurements for a glutamine mutant engineered individually at each N-linked glycan residue in the H77 strain were reported in (Goffard et al. 2005). However, glutamine was not observed at these residues in the available sequence data. Thus, for comparing our model predictions with the infectivity of these mutant H77 strains, we used the energies corresponding to the H77 strains with the least-frequent amino acid at the respective residues. A statistically significant high negative correlation was observed between the predicted energies and the reported infectivity of the mutant strains (Fig. R10a). A similar high correlation was also obtained between our model predictions and the infectivity of strains having (unobserved) alanine mutants that were reported in (Rothwangl et al. 2008) (Fig. R10b).

Figure R10 | Logarithm of fitness vs. predicted energy.

Results using experiments reported in (a) (Goffard et al. 2005) and (b) (Rothwangl et al. 2008). Both fitness and energy are normalized with the reference strain (H77) used in the experiment.

We thank the reviewer for mentioning this important point (reviewer 1 also raised a similar point; please refer to the last response of that reviewer for further details). Although we were accounting for these rare mutations in our analysis in the manner described above, we recognize that the related discussion was missing. We have added this detail in the Methods section of the revised manuscript.

2. *In principle, this gives $20 \times 350 = 7000$ single site fitness coefficients and $\sim 20^2 \times 350^2 / 2 \sim 24$ million pair-wise coefficients to be determined from only 3000 sequences - this clearly is impossible. It is not clear what their assumptions are in this regard. In any case they can only determine coefficients in the model for amino acids that are actually observed in this sample - it is possible that they assume anything below their detection threshold is not important - some discussion of this assumption would be good along with some idea of how much variation they are missing, giving known values of $\mu \times N$.*

Response:

We appreciate the reviewer's comment and the potential confusion. These methodological details were deliberately excluded as a detailed discussion was already provided in our previous work which proposed the underlying inference method (Louie et al. 2018). Nonetheless, we discuss these below and have now included further methodological details in the revised manuscript for completeness, as suggested by the reviewer in the "Summary" comment.

Regarding the number of parameters in our model, we indeed do not include all 24 million pair-wise coefficients. Instead, for each residue, we only model a reduced subset of the amino acids at that residue. To find what this reduced subset is, we combine the least frequently observed amino acids from the sequence data together. Specifically, if we only observe $q_i + 1$ amino acids at residue i in the sequence data, then we only model the $k_i + 1$ most frequently observed amino acids, while the remaining $q_i - k_i$ amino acids are grouped together. To determine the value of k_i , we consider an approach which tries to introduce as much bias due to amino acid grouping as possible within the limits afforded by the fluctuations in the amino acid frequencies. This is to prevent overfitting, with the additional advantage of reduced computational complexity. We also do not need to consider the wildtype (most frequently occurring) amino acid in the model, as this will lead to overparameterization (as the sum of the amino acid frequencies is equal to one). Finally, we also include L1 regularization in our optimization algorithm to infer the parameters, which has the effect of inducing further sparsity in the model. Sparsity is a reasonable assumption as the majority of residues are far apart in the crystal structure, and thus are less likely to have strong interaction. These methodological details are now included in the "Methods" section in the main text.

Note that inferring a maximum-entropy model from the data, as discussed above, has been argued to be meaningful only when the underlying problem is "well-conditioned", i.e., a small change in the data only affects very few model parameters (Cocco & Monasson 2012). This is equivalent to imposing a sparsity constraint on the inferred interaction network. This assumption is valid for certain problems in which the interaction network being inferred is known to be sparse, e.g., atomic spins in a two-dimensional ferromagnet interact only with their neighbors, the tertiary structure of a protein is formed by interaction between very few residues, etc. However, such a priori information may not be available for all problems and thus the inferred sparse model in these cases needs to be validated. In the present work, we validated our inferred model by comparing its predictions with the available biological data, which demonstrate excellent agreement (Fig. 2). We note that multiple studies, inspired by the inference method proposed in (Cocco & Monasson 2012), also used the sparsity assumption on interacting residues for inferring fitness landscapes of viral proteins (Mann et al. 2014; Barton et al. 2016), and associated comparisons with in vitro fitness measurements and in vivo viral evolution in these papers has supported the validity of these assumptions. Moreover, maximum-entropy-based models used for predicting protein contacts (Morcos et al. 2011; Cocco et al. 2013; De Leonardis et al. 2015; Jacquin et al. 2016; Cocco et al. 2018); and coevolution-based methods for identifying protein functional domains (Halabi et al. 2009; Dahirel et al. 2011; Novinec et al. 2014; Rivoire et al. 2016; Salinas & Ranganathan 2018) provide a wealth of evidence that while residue-residue interactions are very important, the strong interactions are generally very sparse.

We also agree that our method "*can only determine coefficients in the model for amino acids that are actually observed in this sample*". We assume that the unobserved amino acid variant may be at least as deleterious as the least-frequent amino acid at the respective residue. Further details and justification of this assumption are provided in the first response.

Finally, it is difficult to get a meaningful measure of how much variation is missing in our model (or in any similar maximum-entropy-based model used in previous studies (Ferguson et al. 2013;

Shekhar et al. 2013; Mann et al. 2014; Barton et al. 2015; Hart & Ferguson 2015; Barton et al. 2016; Flynn et al. 2017; Louie et al. 2018)), particularly since the sequences are not samples of a simple evolutionary process, nor are they taken at any evolutionary equilibrium. While estimates of the effective population size ($N_e \sim 2000$ (Bull et al. 2011)) and mutation rate for HCV ($\mu \sim 1$ mutation/genome/replication cycle (Geller et al. 2016)) have been specified; building a model for quantifying the missing variation that truly reflects the HCV evolutionary dynamics—shaped by multiple factors that include host-specific but population-diverse immunity, reversion, genetic drift, recombination, etc.—is extremely non-trivial.

While it is difficult to quantify precisely how much variation we are missing, we can demonstrate that our model is at least robust to missing variation in the data. Specifically, we can show that the statistics used to train our model—the single and double mutant probabilities—are robust to the data, and are not significantly affected by the addition or removal of particular sequences in the MSA. This was tested by recomputing the single and double mutant probabilities for a subsampled version of the MSA and comparing them with those obtained using the complete MSA. The median of the normalized root-mean-square-error (NRMSE) of the probabilities calculated using the subsampled MSA converged to within 0.05, even when only half of the total sequences were used to construct the subsampled MSA (Fig. R11). This result suggests that the variation missing in the data may not result in drastic changes in the reported predictions, since our model is based purely on the first and second order mutant probabilities. An additional important remark is that, due to the high genetic diversity of HCV E2, it is expected that the lack of observed variation would most probably be highly deleterious (associated with high-fitness-cost; please refer to the first response). Hence, missing such variations may not be important for the main application of our model, i.e., identification of E2 residues with *low-fitness-cost* (*easy escape*) mutations that potentially assist in escape from HmAbs.

We have incorporated comments about the robustness of the single and double mutant probabilities in the Methods section and have included Fig. R11 as Supplementary Fig. S8.

Figure R11 | Robustness of the (a) single and (b) double mutation probabilities to the number of sequences in the MSA.

Each box plot shows the normalized root-mean-square-error NRMSE of the probabilities observed in the subsampled (sampling with replacement) MSA for 500 runs, where B is the total number of sequences (after data preprocessing). It can be observed that the median of the NRMSE of the probabilities calculated using the subsampled MSA converged to within 0.05, even when only half of the total sequences were used to construct the subsampled MSA.

3. A correlation of 0.74 is good, but given that even triple mutant probabilities are reproduced (which suggests triple mutant probabilities are products of combinations of double and single mutant effects) - so why not a better correlation - what is missing in the model?

Response:

Our model indeed reasonably predicts additional statistics of the MSA—including the triple mutant probabilities (Fig. 1c) and the distribution of the number of mutations (Fig. 1d)—which were not explicitly used to train it. This specifically demonstrates that the inferred maximum entropy model is a good generative statistical model that can reflect the *prevalence* of circulating viral strains in the population. This does not however imply that it will be a good proxy for *viral fitness*; particularly given the complex evolutionary dynamics of HCV. The same is true, in fact, for any virus, and the relationship between prevalence and fitness depends on the specific evolutionary properties; generally in a very complicated manner. For example, in the case of the haemagglutinin protein of the influenza A virus, correspondences between predicted (cross-sectional) prevalence and intrinsic viral fitness are very distinct to that of HCV E2, largely since the evolution is strongly driven by effective herd memory responses (Łuksza & Lässig 2014; Neher et al. 2014). Generally speaking, despite many on-going efforts, there is still much work to be done in gaining a mechanistic understanding of how intrinsic fitness shapes the observed prevalence of circulating viral populations, and how this may be best leveraged to improve the inference of fitness parameters from observed samples.

Nonetheless, for HIV proteins, for a simple monotonic relationship between predicted prevalence and viral fitness to hold, three key factors were previously reported (Shekhar et al. 2013; Zanini et al. 2015): (i) a diverse (and ineffective) immune response due to host genetic diversity; (ii) reversion of deleterious mutations on transmission to a new host; and (iii) absence of an effective natural or vaccine-induced herd memory response. For many HIV proteins, the monotonic correspondence between prevalence and fitness was confirmed by demonstrating a high correlation of predicted prevalence with in vitro fitness measurements as well as in vivo virus evolution studies (Ferguson et al. 2013; Mann et al. 2014; Barton et al. 2016; Louie et al. 2018). While HCV is a different virus, it is similar to HIV in many aspects (Franciscus 2017) including the above-mentioned factors. Specifically, since most HCV sequences are sampled from chronic patients and E2 is immunogenic (Ward et al. 2002), it is also likely subjected to diverse and ineffective immune responses in such patients. Reversion to the consensus amino acid upon transmission to a new host has also been reported for HCV (Timm et al. 2004; Ansari et al. 2017). The above points are supported by the observed high correlation between predicted prevalence of HCV E2 sequences and in vitro fitness measurements (Fig. 2a).

To summarize, due to the complex interplay between the prevalence and fitness (as discussed above), the prevalence predicted by even a perfect statistical model is not expected (a priori) to perfectly correlate with intrinsic viral fitness. However, it is encouraging that we obtain a high correlation. While the points indicated above suggest that a monotonic relationship may exist, the ultimate validation must be via comparisons with experiments. We believe that the observed correlation of 0.74 between prevalence predicted by our model and in vitro viral fitness measurements is quite significant, and note that it is also at par with that observed for HIV proteins and the HCV NS5B protein in previous studies (Ferguson et al. 2013; Mann et al. 2014; Hart & Ferguson 2015; Louie et al. 2018).

We have included the above points in the Discussion section of the revised manuscript.

4.

- A. This leads to the key problem I have with this computational technique, is that this analysis could mask particular combinations of background sequences that are of low fitness cost, despite the overall average at that site being of high fitness cost. Hence the mean itself is potentially meaningless without some idea of the distribution of fitness effects at each site/residue. What is important is something like the minimum fitness cost at each site when examined over all background sequences.

General note on response to comment 4: We thank the reviewer for such in-depth comments related to the proposed computational technique. We have collated multiple related comments to address them together in a logical and efficient manner.

Response to comment A: We agree that averaging can potentially mask low-fitness-cost mutations at a residue in some specific background sequences. However, as we discuss below, our proposed metric ΔE_i is designed such that it **does not** mask the low-fitness-cost mutations that can potentially assist in escaping antibodies. Importantly, these mutations are ones that (i) occur on fit backgrounds, and (ii) do not substantially reduce fitness when they occur.

We first discuss an alternative metric along the lines suggested by the reviewer (i.e., one designed to identify the *lowest-fitness-cost* mutation at each residue), and clarify the issues with this approach for our purposes. We then demonstrate that our proposed metric—which weights the cost of making a mutation in a particular background sequence with the prevalence of that background (equation 10)—is a more suitable metric, and show this does not lead to masking of easy escape pathways from antibodies. Equations 10 and 11 are reproduced here respectively for convenience

$$\Delta E_{i,h,J}(a) = \sum_{\mathbf{x}, x_i=0} (E_{h,J}(\mathbf{x}') - E_{h,J}(\mathbf{x})) p_{h,J}(\mathbf{x}), \quad (10)$$

$$\Delta E_i = \frac{\sum_{a=1}^{q_i} \Delta E_{i,h,J}(a) \exp[-\Delta E_{i,h,J}(a)]}{\sum_{a=1}^{q_i} \exp[-\Delta E_{i,h,J}(a)]}. \quad (11)$$

First, as suggested by the reviewer, we define ΔE_i^{\min} as the metric which, for a given residue i , identifies the lowest-fitness-cost mutation over all the background sequences and all non-consensus amino acids at that residue. Contrasting this against our metric ΔE_i (defined in equations 10 and 11), it can be defined by the analogous equations:

$$\Delta E_{i,h,J}^{\min}(a) = \min_{\mathbf{x}, x_i=0} (E_{h,J}(\mathbf{x}') - E_{h,J}(\mathbf{x})) \quad (R1)$$

$$\text{and } \Delta E_i^{\min} = \min_a \Delta E_{i,h,J}^{\min}(a) \quad (R2)$$

where in equation R1, similar to equation 10, \mathbf{x} includes the background sequences in the generated ensemble having the consensus amino acid ('0') at the i -th residue; and, for any given \mathbf{x} , \mathbf{x}' is defined as the corresponding strain, but with amino acid a at residue i .

We compared ΔE_i^{\min} with our metric ΔE_i for each residue, with the results shown in Fig. R12a. While the trends are broadly similar, there are some instances in which the metric ΔE_i^{\min} is small and ΔE_i is large, which may suggest that our metric is masking some low-fitness-cost mutations, at least for some background sequences. However, the metric ΔE_i^{\min} simply reflects the smallest *relative* fitness cost incurred by making a mutation on some background sequence, and it does not take into account the fitness of the background sequence itself. For this reason, the metric can be misleading.

For example, considering a mutation that results in zero change in energy (or no change in fitness) on a given background, ΔE_i^{\min} may suggest that this is potentially an easy escape mutation against antibodies targeting that specific residue. However, if the background on which this mutation occurred was already very deleterious, then that background is unlikely to arise in the first place, and even if it did, the sequence carrying the mutant would still be unfit. Thus, for a mutation to be categorized as providing an easy escape from antibodies, not only should the change in energy

$(E_{h,j}(\mathbf{x}') - E_{h,j}(\mathbf{x}))$ associated with the mutation should be small (or negative), but the background should also be reasonably fit (i.e., the prevalence $p_{h,j}(\mathbf{x})$ of the background sequence should be high). Our ΔE_i metric (in equation 10) takes this into account by weighting the change in energy by the prevalence of the background sequence. Specifically, the changes in energy for those background sequences with higher prevalence/fitness are given a larger weight.

While incorporating the prevalence of the background sequence is important, one may argue that our metric ΔE_i may still potentially mask low-fitness-cost mutations in prevalent background sequences (high $p_{h,j}(\mathbf{x})$) for which the change in energy is very close to zero ($(E_{h,j}(\mathbf{x}') - E_{h,j}(\mathbf{x})) \approx 0$). To test that this does *not* happen, we defined and analyzed a new metric $\Delta E_i^{\text{min,pre-filtered}}$ that restricts the calculation of the above metric ΔE_i^{min} (equations R1-R2) to only the “reasonably prevalent” sequences, which we define to be those occurring at least twice in the sequence ensemble, and then compared it with the proposed metric ΔE_i over all residues. Denoting these prevalent sequences as φ , this is given by

$$\Delta E_{i,h,j}^{\text{min,pre-filtered}}(a) = \min_{x \in \varphi, x_i=0} (E_{h,j}(\mathbf{x}') - E_{h,j}(\mathbf{x})) \quad (\text{R3})$$

$$\text{and } \Delta E_i^{\text{min,pre-filtered}} = \min_a \Delta E_{i,h,j}^{\text{min,pre-filtered}}(a). \quad (\text{R4})$$

The idea of this new metric $\Delta E_i^{\text{min,pre-filtered}}$ is to filter out the low-fitness-cost mutations that occur on background sequences associated with low prevalence, and thus it provides a more meaningful predictor of potential antibody escape mutations as compared to ΔE_i^{min} . The results of this analysis, reported in Figs. R12b and R12c, demonstrated that unlike in Fig. R12a discussed above, there are now very few residues for which ΔE_i is large while $\Delta E_i^{\text{min,pre-filtered}}$ is small, suggesting that any potential masking of easy escape mutations with our proposed metric is minimal.

Moreover, for the HmAb binding residues (shown in color), there is **no evidence of masking at the 20% binding level** (Fig. R12b), while at the (looser) 40% binding level, possible masking is seen for only a single binding residue. The corresponding HmAb was however already predicted by our metric ΔE_i to be escape-susceptible (Fig. R12c) (i.e., this HmAb involved easy escape mutations at other binding residues as well); thus, even if one uses the metric $\Delta E_i^{\text{min,pre-filtered}}$ in place of ΔE_i , results consistent with those in Figs. 3 and 4 would still be obtained.

The overall conclusion of this analysis is that our approach does not lead to masking of antibodies with easy escape pathways.

It is noteworthy that in addition to weighting changes in energy by background prevalence, another important feature of our metric that protects against masking easy escape pathways is the *non-uniform Boltzmann average* performed over all non-consensus amino acids at a residue i (equation 11). This has the effect of (exponentially) emphasizing low-fitness-cost mutations, and is very similar to selecting the minimum $\Delta E_{i,h,j}(a)$ over all non-consensus amino acids $a = 1, \dots, q_i$ observed at residue i , i.e., replacing equation 11 by $\Delta E_i^{\text{a min}} = \min_a \Delta E_{i,h,j}(a)$ (Fig. R12d).

Figure R12 | Tests related to possible masking of low-fitness-cost mutations in the proposed fitness-cost metric ΔE_i .

(a) Comparison of the proposed metric ΔE_i and the metric ΔE_i^{min} that assigns the minimum-fitness-cost to a mutation at residue i over all the background sequences, irrespective of the prevalence of the background sequence (defined in equations R1-R2). (b-c) Comparison of the proposed metric ΔE_i and the metric $\Delta E_i^{min,pre-filtered}$ that assigns the minimum-fitness-cost to a mutation at a residue i calculated over only the prevalent background sequences (defined in equations R3-R4). The binding residues of the predicted escape-susceptible and escape-resistant HmAbs are shown in blue and red color, respectively. The binding residues are defined based on (b) $RB \leq 20\%$ and (c) $RB \leq 40\%$. (d) Comparison of the proposed metric ΔE_i , that uses the Boltzmann averaging over all the observed non-consensus amino acids at residue i (equation 11); and a new metric $\Delta E_i^{a,min}$, which is defined identically to ΔE_i , but in place of the Boltzmann average (in equation 11), selects the *minimum* weighted change in energy among any observed non-consensus amino acid at residue i .

To summarize, our proposed metric ΔE_i predicts the average fitness cost incurred by the virus upon changing a residue i from the consensus to any mutant amino acid in an ensemble of potentially viable viral strains (i.e., a representative viral quasispecies) typically circulating in the population. While there is some possibility of missing low-fitness-cost mutation in a particular background using this metric; the above tests demonstrate that these are likely to occur only in less prevalent (low fitness) backgrounds and therefore are not likely to constitute easy escape pathways. From a vaccine perspective, our proposed metric seems appropriate as one would like to design a vaccine that elicits escape-resistant antibodies against the strains typically observed in the population.

We are grateful to the reviewer for raising this comment, which deals with a rather deep technical point concerning the metric. We also realize that we were not precise enough in our description of the proposed metric as well as in the definition of easy escape mutations, that may have caused confusion. We have addressed these issues by updating the related text in the Methods and Results section. The above detailed discussion has been included in Supplementary Text S2.

B. *Even if a single mutational path exists of low fitness cost then it is just a question of how long it takes for that path to arise. In practice, if it requires many changes at other sites, then it may not be likely to arise on relevant time scales - however, if it requires a double or triple mutant across other sites then this could be arise quickly given a large mutational input ($N \cdot \mu \gg 1$).*

Response to comment B: We understand the reviewer's concern to be that if a residue has a high ΔE_i , it may still be possible for an individual background sequence \mathbf{x} to contribute a term $(E_{h,j}(\mathbf{x}') - E_{h,j}(\mathbf{x})) p_{h,j}(\mathbf{x})$ having a small value in the summation in equation 10, which may in turn lead to "masking" of potential easy-escape pathways, and for such pathways, one may ask about the time and number of mutations needed for escape to occur. However, as we have detailed in response to comment 4A, masking does not happen, at least when considering the escape-resistant binding residues, which are of primary interest in this paper.

Of course, for antibodies which *are* predicted to have easy escape pathways, one may ask about the associated time and number of mutations needed to escape. We appreciate the interest in such related questions about escape dynamics, but we believe that these are most suitably addressed in a separate independent study. Here we adopt the general perspective that, given the high genetic diversity of HCV, if easy escape pathways exist, then such pathways will eventually be selected. Actually, it has been estimated that all possible single and double mutants are generated multiple times each day in a HCV-infected individual, and escape may occur eventually via sequential mutations if a low-fitness-cost pathway exists (Rong et al. 2010). Thus from a vaccine perspective, this motivates one to ask if there exist HmAbs that are difficult for HCV to escape. We have tried to address this question, rather than looking at specifics about the escape dynamics for those HmAbs that do seem to have easy escape pathways.

C. *So in the figures in Fig.2 which seems to show box plots over regions of the protein, these give an indication of the variance of the mean fitness costs over a number of residues and so also mask potential low fitness cost mutants.*

Response to comment C: As described in the response to comment 4A, our proposed metric does not mask potential low-fitness-cost mutations involved with antibody escape. Thus, the results presented in Fig. 3 (and also Fig. 4) do not miss any potential escape mutations. In fact, as discussed in the response to comment 4A, there is no change in the results presented in Fig. 3 (and Fig. 4) even if one uses a different metric $\Delta E_i^{\text{min,pre-filtered}}$ which is not based on averaging (Figs. R12b and R12c).

D. *To address this there would need to be some analytical or simulation analysis, using their empirically determined fitness function that confirms or not that regions identified as being "free" of low fitness mutations indeed do not develop escape mutations in concert with other changes in the protein. There are various options for the simulations, but they would need to allow there to be polymorphisms in the population - so for example individual based simulations. The tricky question in the simulations is what effective population size should be used. I do not know of any studies of N in HCV, but in HIV it has consistently been measured to be of order $\sim 10^4$ or 10^5 , thought to be reflective of population structure and maybe more simply that only infected and actively replicating cells produce genetic variation, and so the effective population size is of order the number of infected cells.*

Response to comment D: A main purpose of the current study was to distinguish escape-resistant antibodies from escape-susceptible ones by predicting the fitness cost of mutating any E2 residue, based on a simulation of an ensemble of sequences that represent typical circulating strains in a HCV-infected population. An in-host simulation (similar to that suggested by the reviewer), in which one starts with a given founder strain, could in principle help to determine the time or the number of mutants required to escape specific HmAbs that are predicted to comprise easy escape mutations. However, as indicated in our response to comment 4B, such a study deals with finer-level questions about escape dynamics, and is an interesting area to explore in the future.

5. ΔE_i is calculated by summing over all other sequences possible except at focal site i (background sequences) - there must be specific background sequences that contribute most to the average ΔE_i - are they predominantly single effect, or double/triple or higher order mutants?

It is difficult to determine which background sequences with a certain number of mutations (e.g., zero/single/double/triple) will contribute most to the average ΔE_i (equation 10). To see why, consider a simple toy example with two residues, and with two amino acids per residue. The sequences can only be $\mathbf{x}=00, 01, 10$ or 11 . Then for the first residue, we have

$$\begin{aligned}\Delta E_{1,h,J}(a) &= \sum_{\mathbf{x}, x_1=0} (E_{h,J}(\mathbf{x}') - E_{h,J}(\mathbf{x})) p_{h,J}(\mathbf{x}) \\ &= (E_{h,J}(10) - E_{h,J}(00)) p_{h,J}(00) + (E_{h,J}(11) - E_{h,J}(01)) p_{h,J}(01) \\ &= h_1(1) p_{h,J}(00) + (h_1(1) + J_{12}(1,1)) p_{h,J}(01)\end{aligned}$$

Observe that the first term $h_1(1) p_{h,J}(00)$ represents the background sequence with no mutations, while the second term $(h_1(1) + J_{12}(1,1)) p_{h,J}(01)$ represents the background sequence with one mutation. The term which contributes the most is dependent on the inferred parameters, $h_1(1)$, $h_2(1)$ and $J_{12}(1,1)$, which in turn influence the prevalence terms $p_{h,J}(00)$ and $p_{h,J}(01)$. The problem is that the inferred parameters can be either positive or negative, and thus $p_{h,J}(00) > p_{h,J}(01)$ or $p_{h,J}(00) < p_{h,J}(01)$. It is thus not clear whether the background with no mutations or single mutation contributes more to $\Delta E_{1,h,J}(a)$, and hence to ΔE_i .

Extending this simple two residue binary model to a model with hundreds of non-binary residues, it will be very difficult to determine which background sequences with a certain number of mutations contribute more to ΔE_i . In fact, if the landscape is sufficiently rugged, which will occur if the coupling terms are not close to zero (as is the case with our landscape), then there will never be a perfect correlation between prevalence and number of mutations. This is illustrated in Fig. R13a, where we plot the distribution of the Spearman correlation between $g_{i,h,J}(a, \mathbf{x}) = (E_{h,J}(\mathbf{x}') - E_{h,J}(\mathbf{x})) p_{h,J}(\mathbf{x})$ and the number of mutants in strain \mathbf{x} , for all a . We observe that the majority of the correlation values are close to -0.1, implying that there is not a strong correlation between the contribution of strain \mathbf{x} to ΔE_i and the number of mutants in strain \mathbf{x} .

Figure R13. Distribution of Spearman correlation between the number of mutants in strain \mathbf{x} and the contribution of that strain to ΔE_i (i.e., $g_{i,h,J}(a, \mathbf{x})$).

Moreover, in the calculation of the ΔE_i metric, the contribution of each background sequence increases with its prevalence $p_{h,J}(\mathbf{x})$. We take a broad perspective that prevalence characterizes the likelihood or “ease” of a particular background arising in a typical viral ensemble. Attempting to quantify the ease of observing a particular background via a “number of mutants” type of metric

does not seem so natural in our context (or even possible, as discussed in the previous paragraph), and choosing a meaningful reference to measure mutations with respect to is not clear (e.g., using the consensus sequence as a reference sequence, there seemed to be a large number of mutations (~20-35 mutations) in such high-contributing background sequences). We appreciate, however, that such a characterization would be suitable in dynamical evolutionary studies, in which case the number of mutations would be meaningfully defined in terms of some obvious reference, such as a transmitted/founder strain.

6. *DE_i is also discussed in the absence of what is the relevant evolutionary scale of fitness or of their energy measure/fitness cost, which is 1/N, where N is the effective population size*

Response:

We appreciate that fitness is an evolutionary quantity whose effect can be measured with respect to suitable evolutionary time-scales. For example, from a population genetics context, how fitness influences the stochastic dynamics of evolving populations (and thus shapes the prevalence) depends on various properties such as population size, mutation rates, recombination, external selective pressures (e.g., immune effects, which may be time varying), transmission bottlenecks, etc. Moreover, in models describing population dynamics, we understand that the parameters such as fitness, mutation rates and recombination are often calibrated against N , since the population size describes the scale of genetic drift (e.g., it reflects the typical time for a neutral mutation to fix purely by chance).

However, such interpretation of evolutionary time scale is most relevant when one thinks about finer-level questions regarding temporal dynamics—such as how long it may take for an easy escape mutation to arise, for antibodies which possess such pathways—and we believe that while interesting, these are not directly relevant to establishing the core results of our paper, as indicated in our response to comment 4B.

More minor comments:

1. *I could not find discussion of the lower right figure of Fig. 1a in the main text*

Response:

We thank the reviewer for reading the manuscript in detail to point this out. We have now included a separate figure for the statistical validation of our model (Fig. 1) and refer to each subfigure explicitly in the main text.

2. *Their minimum probability flow method should be outlined in the methods and or SI, as it is a new and relatively non-standard optimisation technique.*

Response:

We agree and have now included a more detailed description of the minimum probability flow algorithm in the Methods section of the revised manuscript.

REFERENCES

- Ansari, M.A. et al., 2017. Genome-to-genome analysis highlights the effect of the human innate and adaptive immune systems on the hepatitis C virus. *Nature Genetics*, 49(5), pp.666–673.
- Babcock, G.J. et al., 2014. High-throughput sequencing analysis of post-liver transplantation HCV E2 glycoprotein evolution in the presence and absence of neutralizing monoclonal antibody T. Wang, ed. *PLoS ONE*, 9(6), p.e100325.
- Bailey, J.R. et al., 2015. Naturally selected hepatitis C virus polymorphisms confer broad neutralizing antibody resistance. *Journal of Clinical Investigation*, 125(1), pp.437–447.
- Barton, J.P. et al., 2016. Relative rate and location of intra-host HIV evolution to evade cellular immunity are predictable. *Nature Communications*, 7, p.11660.
- Barton, J.P., Kardar, M. & Chakraborty, A.K., 2015. Scaling laws describe memories of host–pathogen riposte in the HIV population. *Proceedings of the National Academy of Sciences*, 112(7), pp.1965–1970.
- Bull, R.A. et al., 2011. Sequential bottlenecks drive viral evolution in early acute hepatitis c virus infection. *PLoS Pathogens*, 7(9).
- Campo, D.S. et al., 2008. Coordinated evolution of the hepatitis C virus. *Proceedings of the National Academy of Sciences*, 105(28), pp.9685–9690.
- Chung, R.T. et al., 2013. Human monoclonal antibody MBL-HCV1 delays HCV viral rebound following liver transplantation: A randomized controlled study. *American Journal of Transplantation*, 13(4), pp.1047–1054.
- Cocco, S. et al., 2018. Inverse statistical physics of protein sequences: a key issues review. *Reports on Progress in Physics*, 81(3), p.032601.
- Cocco, S. & Monasson, R., 2012. Adaptive Cluster Expansion for the Inverse Ising Problem: Convergence, Algorithm and Tests. *Journal of Statistical Physics*, 147(2), pp.252–314.
- Cocco, S., Monasson, R. & Weigt, M., 2013. From principal component to direct coupling analysis of coevolution in proteins: Low-eigenvalue modes are needed for structure prediction. *PLoS Computational Biology*, 9(8), p.e1003176.
- Cowton, V.M. et al., 2018. Predicting the Effectiveness of Hepatitis C Virus Neutralizing Antibodies by Bioinformatic Analysis of Conserved Epitope Residues Using Public Sequence Data. *Frontiers in Immunology*, 9(June), p.1470.
- Dahirel, V. et al., 2011. Coordinate linkage of HIV evolution reveals regions of immunological vulnerability. *Proceedings of the National Academy of Sciences*, 108(28), pp.11530–11535.
- Ferguson, A.L. et al., 2013. Translating HIV sequences into quantitative fitness landscapes predicts viral vulnerabilities for rational immunogen design. *Immunity*, 38(3), pp.606–617.
- Flynn, W.F. et al., 2017. Inference of epistatic effects leading to entrenchment and drug resistance in HIV-1 protease. *Molecular Biology and Evolution*, 34(6), pp.1291–1306.
- Franciscus, A., 2017. Similarities and Differences between HIV and HCV. *HCSF Advocate*, pp.1–3.
- Geller, R. et al., 2016. Highly heterogeneous mutation rates in the hepatitis C virus genome. *Nature Microbiology*, 1(7), p.16045.
- Goffard, A. et al., 2005. Role of N-linked glycans in the functions of hepatitis C virus envelope glycoproteins. *Journal of Virology*, 79(13), pp.8400–8409.
- Halabi, N. et al., 2009. Protein sectors: Evolutionary units of three-dimensional structure. *Cell*, 138(4), pp.774–86.
- Hart, G.R. & Ferguson, A.L., 2015. Empirical fitness models for hepatitis C virus immunogen design. *Physical Biology*, 12(6), p.066006.
- Jacquin, H. et al., 2016. Benchmarking Inverse Statistical Approaches for Protein Structure and Design with Exactly Solvable Models. *PLoS Computational Biology*, 12(5), p.e1004889.
- de Jong, Y.P. et al., 2014. Broadly neutralizing antibodies abrogate established hepatitis C virus infection. *Science Translational Medicine*, 6(254), p.254ra129–254ra129.
- Keck, Z.-Y. et al., 2005. Analysis of a highly flexible conformational immunogenic domain A in hepatitis C virus E2. *Journal of Virology*, 79(21), pp.13199–13208.
- Kim, A.Y. et al., 2011. Spontaneous control of HCV is associated with expression of HLA-B*57 and preservation of targeted epitopes. *Gastroenterology*, 140(2), pp.686–696.
- De Leonadis, E. et al., 2015. Direct-Coupling Analysis of nucleotide coevolution facilitates RNA secondary and tertiary structure prediction. *Nucleic Acids Research*, 43(21), pp.10444–10455.
- Louie, R.H.Y. et al., 2018. Fitness landscape of the human immunodeficiency virus envelope protein that is targeted by antibodies. *Proceedings of the National Academy of Sciences*, 115(4), pp.E564–E573.
- Łuksza, M. & Lässig, M., 2014. A predictive fitness model for influenza. *Nature*, 507(7490), pp.57–61.

- Mann, J.K. et al., 2014. The fitness landscape of HIV-1 Gag: Advanced modeling approaches and validation of model predictions by in vitro testing R. R. Regoes, ed. *PLoS Computational Biology*, 10(8), p.e1003776.
- McKiernan, S.M. et al., 2004. Distinct MHC class I and II alleles are associated with hepatitis C viral clearance, originating from a single source. *Hepatology*, 40(1), pp.108–114.
- Morcos, F. et al., 2011. Direct-coupling analysis of residue coevolution captures native contacts across many protein families. *Proceedings of the National Academy of Sciences of the United States of America*, 108(49), pp.E1293–E1301.
- Morin, T.J. et al., 2012. Human monoclonal antibody HCV1 effectively prevents and treats HCV infection in chimpanzees J. J. Ou, ed. *PLoS Pathogens*, 8(8), p.e1002895.
- Neher, R.A., Russell, C.A. & Shraiman, B.I., 2014. Predicting evolution from the shape of genealogical trees. *eLife*, 3, pp.1–18.
- Novinec, M. et al., 2014. A novel allosteric mechanism in the cysteine peptidase cathepsin K discovered by computational methods. *Nature Communications*, 5.
- Oniangue-Ndza, C. et al., 2011. Compensatory Mutations Restore the Replication Defects Caused by Cytotoxic T Lymphocyte Escape Mutations in Hepatitis C Virus Polymerase. *Journal of Virology*, 85(22), pp.11883–11890.
- Parera, M. & Martinez, M.A., 2014. Strong epistatic Interactions within a single protein. *Molecular Biology and Evolution*, 31(6), pp.1546–1553.
- Pierce, B.G. et al., 2016. Global mapping of antibody recognition of the hepatitis C virus E2 glycoprotein: Implications for vaccine design. *Proceedings of the National Academy of Sciences*, 113(45), pp.E6946–E6954.
- Rivoire, O., Reynolds, K.A. & Ranganathan, R., 2016. Evolution-Based Functional Decomposition of Proteins. *PLoS Computational Biology*, 12(6), p.e1004817.
- Rong, L. et al., 2010. Rapid Emergence of Protease Inhibitor Resistance in Hepatitis C Virus. *Science Translational Medicine*, 2(30), p.30ra32-30ra32.
- Rosner, B., 1983. Percentage Points for a Generalized ESD Many-Outlier Procedure. *Technometrics*, 25(2), p.165.
- Rothwangl, K.B. et al., 2008. Dissecting the role of putative CD81 binding regions of E2 in mediating HCV entry: Putative CD81 binding region 1 is not involved in CD81 binding. *Virology Journal*, 5(1), p.46.
- Salinas, V.H. & Ranganathan, R., 2018. Coevolution-based inference of amino acid interactions underlying protein function. *eLife*, 7, pp.1–43.
- Shekhar, K. et al., 2013. Spin models inferred from patient-derived viral sequence data faithfully describe HIV fitness landscapes. *Physical Review E*, 88(6), p.062705.
- Singh, R. et al., 2007. A comparative review of HLA associations with hepatitis B and C viral infections across global populations. *World journal of gastroenterology*, 13(12), pp.1770–87.
- Sormani, P., Aprile, F.A. & Vendruscolo, M., 2015. Rational design of antibodies targeting specific epitopes within intrinsically disordered proteins. *Proceedings of the National Academy of Sciences*, 112(32), pp.9902–9907.
- Tietjen, G.L. & Moore, R.H., 1972. Some Grubbs-type statistics for the detection of several outliers. *Technometrics*, 14(3), pp.583–597.
- Timm, J. et al., 2004. CD8 epitope escape and reversion in acute HCV infection. *The Journal of experimental medicine*, 200(12), pp.1593–1604.
- Velázquez-Moctezuma, R. et al., 2017. Applying antibody-sensitive hypervariable region 1-deleted hepatitis C virus to the study of escape pathways of neutralizing human monoclonal antibody AR5A. *PLoS Pathogens*, 13(2), pp.1–29.
- Ward, S. et al., 2002. Cellular immune responses against hepatitis C virus: the evidence base 2002. *Clinical & Experimental Immunology*, 128(2), pp.195–203.
- Zanini, F. et al., 2015. Population genomics of intrapatient HIV-1 evolution. *eLife*, 4, p.e11282.

Reviewers' Comments:

Reviewer #1:

Remarks to the Author:

Most of the comments have been adequately addressed by the authors. Their response to comment 5, which includes a new figure (Fig. 2C) that points to possible compensatory fitness-improving mutations in the context of H77 with the N417S substitution, is quite interesting. However, the authors should clarify whether these identified variants (e.g. Q444R) are specific for N417S based on this analysis, or whether unmutated H77 likewise shows this trend and the same (or other) mutations potentially improving viral fitness. Other extra-epitope variants have been observed by other groups for antibodies targeting this region associated with compensatory fitness effects (e.g. E655G in the context of N415Y for the AP33 mAb, Gal-Tanamy et al. PNAS 105 (49): 19450-19455), and it's not clear whether the specific epitope mutation at N415 or N417 has an effect on this analysis. In this regard, it would be useful for readers if the authors can include a supplemental table listing such fitness-improving variants for H77 (and possibly H77-N417S) and their ΔE 's, for the reference of future studies, given that such non-epitope viral escape mutations are of continued interest in the HCV field. Relatedly, a supplemental table with the numbers listing all of the E2 per-residue fitness costs (represented by the colors in Fig. 3A) would be a useful reference for prospective studies in this area.

Reviewer #2:

Remarks to the Author:

Firstly I should say that because this method is aiming to be published in a high profile journal it carries particular weight, which could lead to important and potentially costly decisions on antibody or vaccine design based on this technique, it is imperative that the method be properly scrutinised and presented with transparency.

I still have strong doubts about the main technique (point 4) and whether it is really catching all low fitness cost escape mutants. I am reasonably happy with the other responses given, except for a technical issue with 5.

The authors are correct that low fitness cost escape mutants arising on rare backgrounds should be discounted, and they are in their approach. But equally, their average overemphasises the high fitness cost mutants that arise on frequent backgrounds, since these have an exponentially small chance of fixing/establishing in the population, which of course is a well-known result from population genetics. What matters is what is the dominant population after some evolutionary process, which the authors metric ignores.

Also although the authors are correct that by Boltzmann weighting the $DE_i(a)$ over amino acids observed is similar to taking a minimum, there is no physical rationale for doing this (the explanation that it selects for the \sim minimum would seem a post-hoc justification in light of my own comments!). On first reading I misunderstood the equations, and assumed this Boltzmann weighting was just the Boltzmann weighting over background sequences, but this is compounded on top of the weighting of background sequences (themselves Boltzmann distributed) - what is the rationale?

Fig.R12 (b&c) is offered as evidence that with a $\min(E)$ metric which is prefiltered to remove uncommon backgrounds, then we are not left with many or any mutations that are masking easy escape mutants despite a large DE_i - this is clearly not true as the grey dots have a broad vertical distribution, indicating a masking of low fitness cost mutants - interpreting this in terms of Fig.3a, in other words, some of the sites marked red or white would turn blue under this filtered metric. It is

true that if we only consider HmAb binding sites this distribution is not as broad, but the analysis is not special to only those sites that current HmAbs bind, and makes predictions about other potential sites, and so it is a weak argument to use these as support for the method. Overall, I find Fig.R12 b&c to be a worrying result that casts doubt on the general accuracy of their method.

Fig.R12 is also confusing, since many HmAb binding sites marked as escape susceptible in blue have large DE_i - this whole discussion has revolved around large DE_i (presuming the metric is doing its job) corresponding to sites free of escape mutants - yet these have been marked susceptible. And apparently in the same figure there are red dots "escape-resistant" mutants which have the same or even lower DE_i than escape susceptible mutants!! This also highlights that the authors have nowhere in the manuscript described exactly how they decide a site is, or is not free, from escape mutants, given a determination their metric DE_i for site i - what threshold is used?

I think there is also an error in their response on page 23 point 5: unless I am mistaken somehow, straightforward substitution gives:

$$dE_{1,h}(a=1) = [h_1(1)-h_1(0) + J_{12}(10) - J_{12}(00)] * p(00) + [h_1(1)-h_1(0) + J_{12}(11) - J_{12}(01)] * p(00)$$

the self-energy term at site 2 must cancel in the difference in energy as that doesn't change in the mutant, but self-energy at site 1 and the interaction terms must remain in full, but the authors have an erroneous expression.

There is finally one other consideration glossed over by the authors, which is that the inferred prevalence landscape is an effective landscape over all sequences in the MSA comprising of the consensus from many different patients collected worldwide at different time points in the progression of the infection and many different environmental factors - on one hand this is an advantage as it captures the broad features that are common across all infections, but on the other it will miss important correlations pertinent to infections in individuals. Understanding the importance of this in determining sites which really are escape resistant in a single patient is not simple, but it is clear that their technique which averages over many patients, may well be masking these also.

To summarise although this approach is important and interesting, claiming to find escape resistant mutants in the E2 envelope, given these large uncertainties I am still not in a position to recommend publication. This is particularly as the authors declined to adopt my recommendation of showing explicitly that their metric works under a range of simulation scenarios - testing thoroughly against simulations, assessing false positives or using ROC plots is standard in the field. The overall impression I get is that the authors believe population genetics and dynamics don't matter for their metric, but to determine whether a protein has no escape mutants requires assessing if out of all possible mutations, an escape mutant arises on some relevant time scale, which involves population genetics!

RESPONSE TO REVIEWERS' COMMENTS

Identifying immunologically-vulnerable regions of the HCV E2 glycoprotein and broadly neutralizing antibodies that target them (NCOMMS-18-11143)

We thank the reviewers once again for the time they have devoted to provide detailed and thoughtful reviews, which have helped to improve our paper. Below, we address each of the reviewers' comments (in blue).

Reviewer #1 (Remarks to the Author):

Most of the comments have been adequately addressed by the authors. Their response to comment 5, which includes a new figure (Fig. 2C) that points to possible compensatory fitness-improving mutations in the context of H77 with the N417S substitution, is quite interesting. However, the authors should clarify whether these identified variants (e.g. Q444R) are specific for N417S based on this analysis, or whether unmutated H77 likewise shows this trend and the same (or other) mutations potentially improving viral fitness. Other extra-epitope variants have been observed by other groups for antibodies targeting this region associated with compensatory fitness effects (e.g. E655G in the context of N415Y for the AP33 mAb, Gal-Tanamy et al. PNAS 105 (49): 19450-19455), and it's not clear whether the specific epitope mutation at N415 or N417 has an effect on this analysis. In this regard, it would be useful for readers if the authors can include a supplemental table listing such fitness-improving variants for H77 (and possibly H77-N417S) and their deltaE's, for the reference of future studies, given that such non-epitope viral escape mutations are of continued interest in the HCV field. Relatedly, a supplemental table with the numbers listing all of the E2 per-residue fitness costs (represented by the colors in Fig. 3A) would be a useful reference for prospective studies in this area.

We thank the reviewer for the positive response. The comments provided have helped to improve our manuscript.

Regarding the fitness-improving mutations in the H77 strain with the N417S substitution (H77_{N417S}), we would like to clarify that the majority of these mutations are not specific to the H77_{N417S} background sequence; they also improve the fitness of the unmutated H77 sequence. We have now listed all fitness-improving mutations in both the H77 and H77_{N417S} background strains in Supplementary Table S1.

As for mutation E655G in the context of the H77_{N415Y} strain (Gal-Tanamy et al. 2008), we did not observe the mutant Tyrosine (Y) at residue 415 in our data. Assuming that the effect of mutation Y would be similar to that of the least-frequent mutant (Aspartic acid, D) observed at this residue, we used our model to identify the effect of all single mutants in the H77_{N415D} strain. Based on this analysis, we did not identify E655G as a fitness-improving mutation in the context of the H77_{N415Y} strain. All fitness-improving mutants were similar to those observed in the context of the H77 strain (listed in Supplementary Table S1).

In addition, we point out that, in response to comments raised by Reviewer 2, we have revised the paper to replace the previous coarse-grained DE_i (deltaE) metric with an evolutionary metric that more explicitly quantifies the ease of escaping immune pressure, for each residue. This metric quantifies the "escape time" from immune pressure directed at each residue, defined as the minimum number of generations of a suitable evolutionary model for mutations at the residue to reach a frequency > 0.5. The model incorporates the inferred fitness landscape to quantify the fitness of viral sequences (see the description in the Supplement – "Population genetics evolutionary model").

The escape time metric correlates strongly with the previous DE_i metric (Fig. R1), and the key messages in the paper remain the same as before (see Figs. 3 and 4). The one exception is the results pertaining to domain E escape mutations. Specifically, we had previously predicted that domain E is escape-resistant based on DE_i. However, the refined escape time metric indicates that while it is relatively difficult to escape domain E (Fig. 3b, *top panel*), there still exist multiple residues at which escape may arise after a reasonably short time (Fig. 3b, *bottom panel*). In fact, these residues include 415 and 417, which is in line with the escape from domain E observed in multiple

reports mentioned by the reviewer (Morin et al. 2012; Gal-Tanamy et al. 2008; Chung et al. 2013; Babcock et al. 2014). In regard to the results related to comparison of our model predictions with those based on a conservation-only model (queried by the reviewer in the previous review round), while the predictions of both models are quite similar for antigenic domains, they exhibit important differences for antibodies, as with the previous metric (included in Supplementary Text S1). In addition, we have included 20 additional fitness measurements and the correlation of these models with fitness measurements have been updated to reflect this.

Moreover, we agree with the reviewer that all of the E2 per-residue escape times, used in plotting Fig. 3a, can be helpful for further prospective studies. We have included these in Supplementary Data 1.

Figure R1 | Comparison of the new evolutionary-model based metric “escape time, t_e^i ” and the previously-used metric “fitness cost, ΔE_i ”.

Reviewer #2 (Remarks to the Author)¹:

Firstly I should say that because this method is aiming to be published in a high profile journal it carries particular weight, which could lead to important and potentially costly decisions on antibody or vaccine design based on this technique, it is imperative that the method be properly scrutinised and presented with transparency.

I still have strong doubts about the main technique (point 4) and whether it is really catching all low fitness cost escape mutants. I am reasonably happy with the other responses given, except for a technical issue with 5.

To summarise although this approach is important and interesting, claiming to find escape resistant mutants in the E2 envelope, given these large uncertainties I am still not in a position to recommend publication. This is particularly as the authors declined to adopt my recommendation of showing explicitly that their metric works under a range of simulation scenarios - testing thoroughly against simulations, assessing false positives or using ROC plots is standard in the field. The overall impression I get is that the authors believe population genetics and dynamics don't matter for their metric, but to determine whether a protein has no escape mutants requires assessing if out of all possible mutations, an escape mutant arises on some relevant time scale, which involves population genetics!

We thank the reviewer for the detailed comments and appreciate the concerns raised in regards to the DE_i metric in particular. After giving this further careful consideration, we agree that in contrast to the coarse-grained characterization provided by DE_i, a population-genetics based evolutionary model can provide a more precise quantification of antibody escape costs. We have therefore taken the reviewer's suggestion and developed such an evolutionary model. Based on this model, we have defined a new metric to replace DE_i, which quantifies the ease of escaping immune pressure directed at each residue. Specifically, the metric captures the "escape time", defined as the minimum number of generations for mutations at the residue to reach a frequency > 0.5.

The evolutionary model is a Wright-Fisher-like model in which sequences undergo discrete rounds of mutation, selection and random sampling (genetic drift) in successive generations. The inferred fitness landscape is used to quantify the fitness of viral sequences during the selection step. We assume a population size of $N_e = 2000$, in line with the known effective population size of HCV for in-host evolution (Bull et al. 2011).

To calculate the number of generations for mutations to reach a frequency > 0.5 for a given residue i , we first pick a sequence randomly from the MSA, with the condition that this sequence has no mutations at that residue. This sequence then forms the initial homogenous population. Sequences in this population then undergo a mutation step, where each nucleotide mutates randomly to another nucleotide with probability $\mu = 10^{-4}$, in line with known HCV mutation probabilities (Cuevas et al. 2009; Sanjuan et al. 2010).

After the mutation step, each sequence then undergoes selection based on the parameters of the inferred fitness model. Specifically, sequence \mathbf{x} in the population will survive with probability

$$f_{h,J}(\mathbf{x}) = \frac{g_{h,J}(\mathbf{x})}{\sum_{\mathbf{y}} g_{h,J}(\mathbf{y})}, \quad \text{with} \quad g_{h,J}(\mathbf{x}) = \frac{e^{\beta(\bar{E} - E_{h,J}(\mathbf{x}))}}{1 + e^{\beta(\bar{E} - E_{h,J}(\mathbf{x}))}} \quad (\text{R1})$$

The function $g_{h,J}(\mathbf{x})$ maps predicted energy (ranging from $-\infty$ to $+\infty$) to fitness by smoothly interpolating between 0, for sequences that have much lower fitness than the population average, and 1 for sequences that have much higher fitness than the population average. Note that for $g_{h,J}(\mathbf{x})$, one can use any monotonic mapping from energy to fitness (Rokhsar, Anderson, and Stein 1986). A similar form for $g_{h,J}(\mathbf{x})$ was also used in previous Wright-Fisher like simulation models (Barton et al. 2016; Shekhar et al. 2013; Amitrano, Peliti, and Saber 1989). Similar to ref. (Barton et al. 2016), we

¹ Note that we have grouped related comments from the reviewer together to address them in a systematic way.

estimated $\beta \sim 0.1$ based on the experimental in-vitro fitness measurements and sequence energies (Supplementary Fig. S8), but we note that the qualitative results are not sensitive to the specific choice of β (Supplementary Fig. S9). To account for the effects of immune pressure on residue i , sequences with a wildtype at that residue have their energy increased (i.e., their fitness decreased) by $b = 10$ before calculation of $f_{h,j}(\mathbf{x})$. This value is chosen to be similar to the largest h_j over all j , so that escape confers a selective advantage for mutants at residue i . Note that choosing other values of b (within $\pm 10\%$ of $b = 10$) leads to similar qualitative results (Supplementary Fig. S9).

As in standard Wright-Fisher models (Ewens 2004), individuals in the next generation are then generated through a multinomial sampling process with parameters N_e and $f_{h,j}(\mathbf{x})$. To predict the number of generations for mutations to reach a majority for residue i , we repeat the above evolutionary steps until the frequency of mutations at residue i is over 0.5. We repeat this procedure using 25 transmitted/founder (T/F) sequences from the MSA, and for each T/F sequence we perform 100 simulation runs. We then average the results to give the final escape time score t_e^i for residue i . The evolutionary model and the escape time metric are described in the Methods section, "Population genetics evolutionary model".

We choose an average prediction measure in order to quantify the typical time for escape from an antibody targeting a specific residue. For a given residue, a higher value of the proposed escape time metric is indicative that antibody pressure directed towards that residue is generally more difficult to escape. We emphasize however that a high value of this metric does not preclude the existence of specific evolutionary scenarios (e.g., specific trajectories originating from specific T/F sequences) for which the escape time is fairly short, but these are expected to occur with very low probability. Moreover, there is an enormous number of possible trajectories a population can take, even for a single T/F strain, and it is extremely difficult to extensively sample these and quantify the shortest possible escape route, even if one wished to do so. Nonetheless, our key objective is to give an estimate of the efficacy of *typical* antibody responses in practice, and the average escape time is more relevant for this purpose.

Additionally, with the proposed escape time measure, the key messages of the paper have remained unchanged. The one exception is the results pertaining to domain E escape mutations. Specifically, we had previously predicted that domain E is escape-resistant based on DE_i . However, the refined escape time metric indicates that while it is relatively difficult to escape domain E (Fig. 3b, *top panel*), there still exist multiple residues at which escape may arise after a reasonably short time (Fig. 3b, *bottom panel*). In fact, these residues include 415 and 417, which is in line with the escape from domain E observed in multiple reports (Morin et al. 2012; Gal-Tanamy et al. 2008; Chung et al. 2013; Babcock et al. 2014).

By replacing the DE_i measure with a more refined escape time metric based on a population genetics model of viral evolution, we believe that most of the previous concerns that were related specifically to issues with the DE_i measure are no longer relevant. Nevertheless, we have collated these below and included our response to the related comments. One review comment that is an exception to this, which appears to be not directed specifically at DE_i , is the following:

There is finally one other consideration glossed over by the authors, which is that the inferred prevalence landscape is an effective landscape over all sequences in the MSA comprising of the consensus from many different patients collected worldwide at different time points in the progression of the infection and many different environmental factors - on one hand this is an advantage as it captures the broad features that are common across all infections, but on the other it will miss important correlations pertinent to infections in individuals. Understanding the importance of this in determining sites which really are escape resistant in a single patient is not simple, but it is clear that their technique which averages over many patients, may well be masking these also.

Our fitness landscape is indeed based on cross-sectional sequence data taken from many individuals at many different time points etc. For reasons that we point out in the discussion (shown previously in the context of HIV also), it most naturally approximates the "intrinsic" fitness landscape of the virus by effectively averaging out diverse immune responses across many individuals. Specifically, using a population genetics Wright-Fisher like model, (Shekhar et al. 2013) previously demonstrated that the inferred fitness landscape (based on a maximum entropy framework) is a good approximation to intrinsic fitness, even when the MSA comprises of the consensus collected from different patients at

different time points, and under different immune pressure. The ability of the model to predict intrinsic fitness is best corroborated by comparison with independent replicative fitness experiments, as we show in Fig. 2a. Nonetheless, we agree that this model does not say anything about specific correlations that may arise in the viral evolution in an individual patient. This requires evolutionary modelling.

The evolutionary model in our revised paper is constructed to describe in-host viral evolution by incorporating intrinsic fitness effects (defined via the inferred fitness landscape) together with external immune pressure forces. This indeed provides a more direct and transparent means of capturing host-specific selective pressures and quantifying escape times than the previous coarse-grained DE_i metric. These escape times take into account broad features of the population, which fits into our key objective of giving an estimate of the efficacy of typical antibody responses in practice. We thank the reviewer, once again, for the encouragement to pursue this evolutionary-model inspired metric.

*DE_i related comments: (*Responses are included for the previous DE_i metric for maximum transparency and completeness; but we note that this DE_i has been fully removed in the revised manuscript):*

- a) The authors are correct that low fitness cost escape mutants arising on rare backgrounds should be discounted, and they are in their approach. But equally, their average overemphasises the high fitness cost mutants that arise on frequent backgrounds, since these have an exponentially small chance of fixing/establishing in the population, which of course is a well-known result from population genetics. What matters is what is the dominant population after some evolutionary process, which the authors metric ignores.

We agree. By construction, the DE_i metric assigned a large weight to any mutation that appeared in frequent backgrounds. The rationale for this specific weighting was that these backgrounds were more likely to be observed in the population as compared to others. However, this metric is not particularly transparent, and we agree that by averaging over all frequent backgrounds a low fitness cost mutation appearing in a particular background would have possibly been masked by the general high fitness cost of the same mutation in other frequent backgrounds. We avoid this issue using the new evolution-based “escape time” metric, in which there is only a small chance a high fitness cost mutation will fixate in a relatively short time, even on a frequent background.

- b) Also although the authors are correct that by Boltzmann weighting the $DE_i(a)$ over amino acids observed is similar to taking a minimum, there is no physical rationale for doing this (the explanation that it selects for the ~minimum would seem a post-hoc justification in light of my own comments!). On first reading I misunderstood the equations, and assumed this Boltzmann weighting was just the Boltzmann weighting over background sequences, but this is compounded on top of the weighting of background sequences (themselves Boltzmann distributed) - what is the rationale?

The rationale for using the Boltzmann averaging was to emphasize the small values (low fitness cost mutations in this case), as was done previously in other related papers (Barton et al. 2016; Louie et al. 2018). The Boltzmann weighting was a little less aggressive than simply taking the minimum, but nonetheless, both had similar effect, as we showed.

- c) Fig.R12 (b&c) is offered as evidence that with a $\min(E)$ metric which is prefiltered to remove uncommon backgrounds, then we are not left with many or any mutations that are masking easy escape mutants despite a large DE_i - this is clearly not true as the grey dots have a broad vertical distribution, indicating a masking of low fitness cost mutants - interpreting this in terms of Fig.3a, in other words, some of the sites marked red or white would turn blue under this filtered metric. It is true that if we only consider HmAb binding sites this distribution is not as broad, but the analysis is not special to only those sites that current HmAbs bind, and makes predictions about other potential sites, and so it is a weak argument to use these as support for the method. Overall, I find Fig.R12 b&c to be a worrying result that casts doubt on the general accuracy of their method.

Although the masking of low fitness cost mutations in the HmAb binding sites was minimal, we agree that the metric may have led to some masking for other E2 residues. This is not an issue with the proposed evolution-based metric for quantifying the ease of escaping immune pressure directed at each E2 residue.

- d) Fig.R12 is also confusing, since many HmAb binding sites marked as escape susceptible in blue have large DE_i - this whole discussion has revolved around large DE_i (presuming the metric is doing its job) corresponding to sites free of escape mutants - yet these have been marked susceptible. And apparently in the same figure there are red dots "escape-resistant" mutants which have the same or even lower DE_i than escape susceptible mutants!! This also highlights that the authors have nowhere in the manuscript described exactly how they decide a site is, or is not free, from escape mutants, given a determination their metric DE_i for site i - what threshold is used?

We apologize for causing confusion due to the use of colours in Fig. R12 similar to Fig. 3a's heat map. In Fig. R12, we had marked the set of *all* binding sites associated with predicted escape-susceptible HmAbs (CBH-4D, CBH-4G, CBH-4B, CBH-20, CBH-21, CBH-22, HC-11, A27, HC84-24, HC84-26, and HC33-4; for $RB \leq 20\%$) and escape-resistant HmAbs (HC-1, CBH-23, CBH-7, HC84-20, and HC33-1; for $RB \leq 20\%$) in blue and red, respectively. As observed in the boxplot in Fig. 4a (of the previous manuscript), the predicted escape-susceptible HmAbs did not only include small DE_i sites but included a large number of sites with large DE_i . Thus, we would expect to see a large number of sites in blue with large DE_i . However, we observed in Fig. R12 that none of the red sites (associated with seemingly escape-resistant HmAbs) had DE_i close to zero, which implied that the DE_i metric was not masking easy escape mutations in the binding sites of HmAbs. However, as mentioned in response to comment c above, some masking was possible for other E2 residues.

As for the definition of sites with easy escape mutations, we had mentioned on page 5 (first line of 2nd paragraph in the previous version of the manuscript) that these are the ones which have $DE_i \sim 0$. However, we recognize that we were not completely explicit in our definition, which may have caused confusion.

- e) I think there is also an error in their response on page 23 point 5: unless I am mistaken somehow, straightforward substitution gives:

$$dE_{1,hJ}(a=1) = [h_1(1) - h_1(0) + J_{12}(10) - J_{12}(00)] * p(00) + [h_1(1) - h_1(0) + J_{12}(11) - J_{12}(01)] * p(00)$$

the self-energy term at site 2 must cancel in the difference in energy as that doesn't change in the mutant, but self-energy at site 1 and the interaction terms must remain in full, but the authors have an erroneous expression.

Any parameter term involving the WT amino acid (i.e., $h(0)$, $J(10)$, or $J(01)$) has a value of zero in our model, as it is the reference. Thus, substituting zero for all terms involving WT amino acids in the equation mentioned by the reviewer, we end up with the equation that we had in our response; specifically, $dE_{1,hJ}(a=1) = [h_1(1)] * p(00) + [h_1(1) + J_{12}(11)] * p(01)$.

References

- Amitrano, C., L. Peliti, and M. Saber. 1989. "Population Dynamics in a Spin-Glass Model of Chemical Evolution." *Journal of Molecular Evolution* 29 (6): 513–25. <https://doi.org/10.1007/BF02602923>.
- Babcock, Gregory J., Sowmya Iyer, Heidi L. Smith, Yang Wang, Kirk Rowley, Donna M. Ambrosino, Phillip D. Zamore, Brian G. Pierce, Deborah C. Molrine, and Zhiping Weng. 2014. "High-Throughput Sequencing Analysis of Post-Liver Transplantation HCV E2 Glycoprotein Evolution in the Presence and Absence of Neutralizing Monoclonal Antibody." Edited by Tianyi Wang. *PLoS ONE* 9 (6): e100325. <https://doi.org/10.1371/journal.pone.0100325>.
- Barton, John P, Nilu Goonetilleke, Thomas C Butler, Bruce D Walker, Andrew J McMichael, and Arup K Chakraborty. 2016. "Relative Rate and Location of Intra-Host HIV Evolution to Evade Cellular Immunity Are Predictable." *Nature Communications* 7 (May): 11660. <https://doi.org/10.1038/ncomms11660>.
- Bull, Rowena A., Fabio Luciani, Kerensa McElroy, Silvana Gaudieri, Son T. Pham, Abha Chopra, Barbara Cameron, et al. 2011. "Sequential Bottlenecks Drive Viral Evolution in Early Acute Hepatitis c Virus Infection." *PLoS Pathogens* 7 (9). <https://doi.org/10.1371/journal.ppat.1002243>.
- Chung, R. T., F. D. Gordon, M. P. Curry, T. D. Schiano, S. Emre, K. Corey, J. F. Markmann, et al. 2013. "Human Monoclonal Antibody MBL-HCV1 Delays HCV Viral Rebound Following Liver Transplantation: A Randomized Controlled Study." *American Journal of Transplantation* 13 (4): 1047–54. <https://doi.org/10.1111/ajt.12083>.
- Cuevas, J. M., F. Gonzalez-Candelas, A. Moya, and R. Sanjuan. 2009. "Effect of Ribavirin on the Mutation Rate and Spectrum of Hepatitis C Virus In Vivo." *Journal of Virology* 83 (11): 5760–64. <https://doi.org/10.1128/JVI.00201-09>.
- Ewens, Warren J. 2004. *Mathematical Population Genetics*. Vol. 27. Interdisciplinary Applied Mathematics. New York, NY: Springer New York. <https://doi.org/10.1007/978-0-387-21822-9>.
- Gal-Tanamy, Meital, Z.-Y. Keck, MinKyung Yi, Jane A McKeating, Arvind H Patel, Steven K H Fong, and Stanley M Lemon. 2008. "In Vitro Selection of a Neutralization-Resistant Hepatitis C Virus Escape Mutant." *Proceedings of the National Academy of Sciences* 105 (49): 19450–55. <https://doi.org/10.1073/pnas.0809879105>.
- Louie, Raymond H Y, Kevin J Kaczorowski, John P Barton, Arup K Chakraborty, and Matthew R. McKay. 2018. "Fitness Landscape of the Human Immunodeficiency Virus Envelope Protein That Is Targeted by Antibodies." *Proceedings of the National Academy of Sciences* 115 (4): E564–73. <https://doi.org/10.1073/pnas.1717765115>.
- Morin, Trevor J, Teresa J Broering, Brett A Leav, Barbra M Blair, Kirk J Rowley, Elisabeth N Boucher, Yang Wang, et al. 2012. "Human Monoclonal Antibody HCV1 Effectively Prevents and Treats HCV Infection in Chimpanzees." Edited by Jing-hsiung James Ou. *PLoS Pathogens* 8 (8): e1002895. <https://doi.org/10.1371/journal.ppat.1002895>.
- Rokhsar, D. S., P. W. Anderson, and D. L. Stein. 1986. "Self-Organization in Prebiological Systems: Simulations of a Model for the Origin of Genetic Information." *Journal of Molecular Evolution* 23 (2): 119–26. <https://doi.org/10.1007/BF02099906>.
- Sanjuan, R., M. R. Nebot, N. Chirico, L. M. Mansky, and R. Belshaw. 2010. "Viral Mutation Rates." *Journal of Virology* 84 (19): 9733–48. <https://doi.org/10.1128/JVI.00694-10>.
- Shekhar, Karthik, Claire Ruberman, Andrew Ferguson, John P. Barton, Mehran Kardar, and Arup Chakraborty. 2013. "Spin Models Inferred from Patient-Derived Viral Sequence Data Faithfully Describe HIV Fitness Landscapes." *Physical Review E* 88 (6): 062705. <https://doi.org/10.1103/PhysRevE.88.062705>.

Reviewers' Comments:

Reviewer #2:

Remarks to the Author:

My main objections on the previous version of the manuscript were that their metric DE_i would overemphasise mutants with low probability of escape which happened to occur on frequent backgrounds and that a more rigorous statistical analysis should be performed. For this reason I suggested a metric related to the time to establishment instead and that false positives be assessed.

In this draft they have taken up my first suggestion and implemented a metric based on the time to escape.

My main remaining concerns are

1) that this new metric is an average escape time over different MSA background sequences rather than the minimum. This is effectively the same objection I had in the first draft applied to this new metric!

2) Only 25 MSA background sequences out of the possible 3000 are used in the simulations to assess the escape time. I also find this approach odd as they have learnt information about all 3000 sequences in their effective energy/fitness function, and this approach discards this information.

3) There is no discussion of what constitutes a sufficiently large escape time to be considered free of escape mutants (i.e. in a statistical sense (see 4) and biologically in terms of the time it takes for antibody binding to have effect and effectively treat the patient).

4) (Related to 3) I recommended that they do a proper statistical analysis including assessing false positives and /or calculating ROC plots (here it would be wrt the escape time), which they have chosen *not* to do or have not addressed directly in their rebuttal.

The authors attempt to answer 1) in their reply by arguing that they calculate an average escape time since they are only looking to assess whether a site with "high probability" has no escape mutants. However, they admit themselves that there may well be sites which they have classified as being escape resistant and yet have accessible mutants with short escape times. This is bordering on double think - how do you know something is without an escape mutant with high probability unless you quantify that probability (e.g. false positive rate - my point 4 above).

However, on balance Fig.2c gives implicit information about the false positive rate and is reasonably convincing empirical evidence that escape times greater than about 200 generations should be a good indication of a site free of escape mutants. However, a more rigorous statistical analysis would be preferable, for example a ROC plot of sites used in Fig.2c ordered by their calculated escape time.

Specific points:

*The authors use a non-standard approach for mapping their energy to survival probability/mean frequency per generation of sequence x - it involves a free parameter β which they calibrate against in vitro experiments, but details of these experiments are not given - did the authors perform the experiments themselves for this paper, if not who did and this work should be referenced?

*An effective population size of $N_e=2000$ is used in the simulations to assess escape time. These estimates tend to have larger (of order magnitude) uncertainty, so the authors should repeat the simulations for $N_e=20000$ to check their results are unchanged.

* The number of replicates used in simulations for each MSA sequence is not written in the Methods, but was given in the reply to be 100 - this should be added to the Methods.

To summarise the following things still need addressing:

* the authors should assess how robust their method is to the small number of MSA background sequences used, by repeating with say 50. They should also compare against randomly drawing background sequences from the eqm distribution for their inferred energy/fitness landscape - it is well-known the equilibrium distribution of sequences is Boltzmann distributed for $\mu^*N \ll 1$ (Sella & Hirsh PNAS, 2005, Iwasa JTB 1988).

* Address point 3 above in Discussion

* for transparency add in discussion issue regarding point 1 above and their choice to use the average time.

*address the specific points above (beta and repeating sims at $N_e=2e4$)

Finally, I have mostly restrained from commenting on novelty and impact as it is outside my expertise to comment on how important it is to find these escape resistant sites in the field of HCV research. However, it is clear that there are a number of very similar papers that have been previously published using the same technique and the calculation of escape time follows exactly the approach in JP Barton,... AK Chakraborty, Nat Comm 2016, so this work is clearly not novel from a methodological/computational viewpoint.

RESPONSE TO REVIEWER'S COMMENTS

Identifying immunologically-vulnerable regions of the HCV E2 glycoprotein and broadly neutralizing antibodies that target them (NCOMMS-18-11143)

Reviewer #1 (Remarks to the Author):

My main objections on the previous version of the manuscript were that their metric DE_i would overemphasise mutants with low probability of escape which happened to occur on frequent backgrounds and that a more rigorous statistical analysis should be performed. For this reason I suggested a metric related to the time to establishment instead and that false positives be assessed.

In this draft they have taken up my first suggestion and implemented a metric based on the time to escape.

We thank the reviewer once again for the time he/she has devoted to provide detailed and thoughtful reviews, which have helped to improve our paper further. Below, we address each of the comments (in blue).

My main remaining concerns are

1) that this new metric is an average escape time over different MSA background sequences rather than the minimum. This is effectively the same objection I had in the first draft applied to this new metric!

*for transparency add in discussion issue regarding point 1 above and their choice to use the average time.

We believe that for characterizing a typical antibody response, an average escape time based on the proposed metric is reasonable. As suggested, we have added comments about this point in the Discussion section of the manuscript. Specifically, we have included the following:

For each E2 residue, the escape time was obtained by averaging over the time it takes to observe escape in a fixed number of transmitted/founder (T/F) sequences (see Methods). We choose this average prediction measure in order to quantify the typical time for escape from an antibody targeting a specific residue. For a given residue, a higher value of the proposed escape time metric is indicative that antibody pressure directed towards that residue is generally more difficult to escape. We emphasize however that a high value of this metric does not preclude the existence of specific evolutionary scenarios (e.g., specific trajectories originating from specific T/F sequences) for which the escape time is fairly short, but these are expected to occur with very low probability. Moreover, there is an enormous number of possible trajectories a population can take, even for a single T/F strain, and it is extremely difficult to extensively sample these and quantify the shortest possible escape route. Nonetheless, the key objective in this work is to give an estimate of the efficacy of typical antibody responses in practice, and the average escape time appears more relevant for this purpose.

2) Only 25 MSA background sequences out of the possible 3000 are used in the simulations to assess the escape time. I also find this approach odd as they have learnt information about all 3000 sequences in their effective energy/fitness function, and this approach discards this information.

*The authors should assess how robust their method is to the small number of MSA background sequences used, by repeating with say 50. They should also compare against randomly drawing background sequences from the eqm distribution for their inferred energy/fitness landscape - it is well-known the equilibrium distribution of sequences is Boltzmann distributed for $\mu^*N \ll 1$ (Sella & Hirsh PNAS, 2005, Iwasa JTB 1988).

We chose to run the evolutionary simulations for only 25 background sequences as these simulations were extremely computationally expensive. This is because for each E2 residue and each transmitted/founder (T/F) sequence, we ran 100 evolutionary simulations for a large number of generations until escape was observed at the studied residue. Conducting these tests for all E2

residues and for multiple T/F strains therefore required enormous computational resources. To deal with this, we ran the simulations on a supercomputer with massive parallel processing capabilities. Using this supercomputer, we have conducted the further robustness tests as suggested by the reviewer, described below, which took **one million core-hours** of processing time (see Supplementary Figs. S9 and S10).

For evaluating the robustness of our results to the number of background or T/F sequences, we repeated the evolutionary simulations with 75 randomly selected (but distinct from the previous 25) T/F sequences from the MSA for the set of critical E2 residues involved in antigenic domains (Fig. 3b) and in forming the binding residues of HmAbs (Fig. 4a). These tests revealed a very strong positive correlation ($r = 0.99$) with the escape time scores computed with the original T/F sequences, providing verification of the robustness of the results to the number of T/F sequences (Fig. R1). In the manuscript, we included the results of these tests, but with a slightly different presentation. Specifically, we partition the 75 T/F sequences into three distinct sets of 25 sequences, and show the variation observed in the computed escape time scores when the evolutionary simulations are repeated for each case (Fig. R2). The message of robustness is again clearly evident from this presentation.

As also suggested, we repeated the simulations by selecting T/F sequences randomly from the inferred model (equilibrium sampling via MCMC simulation). The escape time scores for the critical E2 residues obtained in this case are also qualitatively very similar to those obtained using T/F sequences from the MSA (Fig. R3), providing yet further validation of the robustness of our results to the set of T/F founder sequences employed.

Figure R1. Robustness of escape time scores to the number of T/F sequences used in the evolutionary simulations.

Figure R2. Robustness of escape time scores to different sets of 25 T/F sequences used in the evolutionary simulations. The y-axis represents the escape times averaged over three randomly selected mutually distinct sets of 25 T/F sequences, with red error bars denoting one standard deviation.

Figure R3. Robustness of escape time scores to the source of T/F sequences (MSA or model) used in the evolutionary simulations.

3) There is no discussion of what constitutes a sufficiently large escape time to be considered free of escape mutants (i.e. in a statistical sense (see 4) and biologically in terms of the time it takes for antibody binding to have effect and effectively treat the patient).

4) (Related to 3) I recommended that they do a proper statistical analysis including assessing false positives and /or calculating ROC plots (here it would be wrt the escape time), which they have chosen *not* to do or have not addressed directly in their rebuttal.

The authors attempt to answer 1) in their reply by arguing that they calculate an average escape time since they are only looking to assess whether a site with "high probability" has no escape mutants. However, they admit themselves that there may well be sites which they have classified as being escape resistant and yet have accessible mutants with short escape times. This is bordering on double think - how do you know something is without an escape mutant with high probability unless you quantify that probability (e.g. false positive rate - my point 4 above).

However, on balance Fig.2c gives implicit information about the false positive rate and is reasonably convincing empirical evidence that escape times greater than about 200 generations should be a

good indication of a site free of escape mutants. However, a more rigorous statistical analysis would be preferable, for example a ROC plot of sites used in Fig.2c ordered by their calculated escape time.

First, we apologize for not sufficiently addressing the ROC point previously. With the clarification above, we now more precisely understand the reviewer's concern and have addressed it as follows.

In Fig. 2c we had compared the distribution of escape times of a few residues at which escape mutation has been observed experimentally/clinically with that of the remaining residues. A classifier designed using the residues with experimentally/clinically-observed escape mutations as true positive control values and assuming all remaining residues as true negative values achieved an average area under the receiver operating characteristic curve (AUC) of 0.93 (Fig. R4). Due to the imbalanced nature of the two classes, i.e., the number of escape-mutation-associated residues being around 1/5th that of the remaining E2 residues, we used the F1 score and the Matthews correlation coefficient (MCC)—two widely used metrics to gauge the performance of binary classifiers with imbalanced classes in the data—to calculate an optimal escape time cut-off value. A similar approach, albeit for a completely different application, was used previously (Yang et al. 2018) to define an interaction score cut-off for identifying protein-protein as well as protein-RNA interactions. For our case, both F1 score and MCC metrics returned the same value of $\zeta = 102$ generations (Fig. R5). Thus, any residue with escape time score more than ζ may be considered potentially difficult to escape and vice versa. We have now revised the results in our manuscript accordingly and we propose only those antibodies as potentially difficult to escape which comprise no binding residue with escape time score less than ζ .

Figure R4. Receiver operating characteristic curve for identifying clinically-observed escape mutations using the inferred escape time scores.

Figure R5. Determination of the optimal escape time cut-off based on the F1 score and Matthews correlation coefficient (MCC). Note that in this binary classification, all residues at which escape mutation is not experimentally/clinically observed are considered difficult to escape.

Regarding the time it takes for antibodies to act and effectively neutralise the virus, this is difficult to answer explicitly due to the complexity of the adaptive processes underlying viral and antibody coevolution (including the complex affinity maturation process underpinning antibody development). Nonetheless, we believe that a scientifically reasonable way to approach this, as also suggested by the reviewer, is to look specifically at escape times that have been observed experimentally or clinically. This is precisely what is done in the statistical analysis indicated above, which suggested that an escape time of roughly 100 generations is a reasonable cut-off in our model.

5) The authors use a non-standard approach for mapping their energy to survival probability/mean frequency per generation of sequence x - it involves a free parameter β which they calibrate against in vitro experiments, but details of these experiments are not given - did the authors perform the experiments themselves for this paper, if not who did and this work should be referenced?

We appreciate the reviewer's comment and the potential confusion. The approach that we used for relating the survival probability of a sequence in the evolutionary simulations to the inferred energy from our model has been previously used in (Barton et al. 2016) to study HIV evolution using experimental infectivity measurements from HIV proteins. We adapted this measure for HCV by using E2 experimental infectivity measurements. Specifically, the involved free parameter " β " was set according to the slope of the weighted linear least-squares (LS) fit between the experimental E2 infectivity measurements reported by different research labs (Goffard et al. 2005; Drummer et al. 2006; Falkowska et al. 2007; Rothwangl et al. 2008; Gal-Tanamy et al. 2008; Z.-Y. Keck et al. 2009; Z. Keck et al. 2012; Guan et al. 2012; Pierce et al. 2016; Gopal et al. 2017; El-Diwany et al. 2017) and the corresponding energies predicted by our model. In the quoted experimental reports, multiple H77 strain (GenBank accession no. NC_004102) variants were constructed using site-directed mutagenesis and the infectivity of each variant was measured. The weighting applied in the LS fit provided calibration in accordance with the number of infectivity measurements in each report (see Methods for details).

We now properly reference these reports in the relevant text in the Methods section to make this clear.

6) An effective population size of $N_e=2000$ is used in the simulations to assess escape time. These estimates tend to have larger (of order magnitude) uncertainty, so the authors should repeat the simulations for $N_e=20000$ to check their results are unchanged.

We have repeated the simulations with effective population size of $N_e = 20000$ and the results are robust to the specific value of N_e (Fig. R7). We restricted these simulations to only 10 T/F sequences due to the enormous computational complexity at this scale. (These simulations were taking ~5 days on average—as compared to ~5 hours for $N_e = 2000$ —to obtain escape time scores for only a single residue and for one T/F sequence.)

Figure R7. Robustness of escape time scores to the value of effective population size (N_e) used in the evolutionary simulations.

7) The number of replicates used in simulations for each MSA sequence is not written in the Methods, but was given in the reply to be 100 - this should be added to the Methods.

We state this in Methods of the revised paper as follows:

We repeat this procedure using 25 T/F sequences from the MSA, and for each T/F sequence we perform 100 simulation runs. We then average the results to give the final escape time score t_e^i for residue i .

To summarise the following things still need addressing:

* the authors should assess how robust their method is to the small number of MSA background sequences used, by repeating with say 50. They should also compare against randomly drawing background sequences from the eqm distribution for their inferred energy/fitness landscape - it is well-known the equilibrium distribution of sequences is Boltzmann distributed for $\mu \cdot N \ll 1$ (Sella & Hirsh PNAS, 2005, Iwasa JTB 1988).

See response to comment 2.

* Address point 3 above in Discussion

See response to comments 3 and 4.

* for transparency add in discussion issue regarding point 1 above and their choice to use the average time.

See response to comment 1.

*address the specific points above (beta and repeating sims at $N_e=2e4$)

See response to comments 5 and 6.

Finally, I have mostly restrained from commenting on novelty and impact as it is outside my expertise to comment on how important it is to find these escape resistant sites in the field of HCV research. However, it is clear that there are a number of very similar papers that have been previously published using the same technique and the calculation of escape time follows exactly the approach in JP Barton,... AK Chakraborty, Nat Comm 2016, so this work is clearly not novel from a methodological/computational viewpoint.

Indeed, the major novelty of our work lies not in method development, but in the application of existing computational methods (developed in different contexts) to reveal novel biological insights related to HCV E2, and in particular, the identification of potentially difficult-to-escape HCV E2-specific antibodies and associated regions. This is the first work that infers a meaningful fitness landscape of HCV E2 and, furthermore, to apply it together with a stochastic population genetics model to quantify the average time to escape from antibody responses targeting any E2 residue. We believe that the identification of antigenic regions/antibodies comprising residues/binding residues of HCV E2 that are potentially difficult to escape from provides significant value which, in particular, can aid the rational design of HCV vaccines.

We have been careful to not claim methodological novelty by giving appropriate references to related prior works in both maximum-entropy inference and evolutionary simulations.

References

- Barton, John P, Nilu Goonetilleke, Thomas C Butler, Bruce D Walker, Andrew J McMichael, and Arup K Chakraborty. 2016. "Relative Rate and Location of Intra-Host HIV Evolution to Evade Cellular Immunity Are Predictable." *Nature Communications* 7 (May): 11660. <https://doi.org/10.1038/ncomms11660>.
- Drummer, H. E., I. Boo, A. L. Maerz, and P. Pombourios. 2006. "A Conserved Gly436-Trp-Leu-Ala-Gly-Leu-Phe-Tyr Motif in Hepatitis C Virus Glycoprotein E2 Is a Determinant of CD81 Binding and Viral Entry." *Journal of Virology* 80 (16): 7844–53. <https://doi.org/10.1128/JVI.00029-06>.
- El-Diwany, Ramy, Valerie J. Cohen, Madeleine C. Mankowski, Lisa N. Wasilewski, Jillian K. Brady, Anna E. Snider, William O. Osburn, Ben Murrell, Stuart C. Ray, and Justin R. Bailey. 2017. "Extra-Epitopic Hepatitis C Virus Polymorphisms Confer Resistance to Broadly Neutralizing Antibodies by Modulating Binding to Scavenger Receptor B1." *PLoS Pathogens* 13 (2): 1–25. <https://doi.org/10.1371/journal.ppat.1006235>.
- Falkowska, E., F. Kajumo, E. Garcia, J. Reinus, and T. Dragic. 2007. "Hepatitis C Virus Envelope Glycoprotein E2 Glycans Modulate Entry, CD81 Binding, and Neutralization." *Journal of Virology* 81 (15): 8072–79. <https://doi.org/10.1128/JVI.00459-07>.
- Gal-Tanamy, Meital, Z.-Y. Keck, MinKyung Yi, Jane A McKeating, Arvind H Patel, Steven K H Fong, and Stanley M Lemon. 2008. "In Vitro Selection of a Neutralization-Resistant Hepatitis C Virus Escape Mutant." *Proceedings of the National Academy of Sciences* 105 (49): 19450–55. <https://doi.org/10.1073/pnas.0809879105>.
- Goffard, Anne, Nathalie Callens, Birke Bartosch, Czeslaw Wychowski, F.-L. Cosset, C. Montpellier, and J. Dubuisson. 2005. "Role of N-Linked Glycans in the Functions of Hepatitis C Virus Envelope Glycoproteins." *Journal of Virology* 79 (13): 8400–8409. <https://doi.org/10.1128/JVI.79.13.8400-8409.2005>.
- Gopal, Radhika, Kelli Jackson, Netanel Tzarum, Leopold Kong, Andrew Ettenger, Johnathan Guest, Jennifer M. Pfaff, et al. 2017. "Probing the Antigenicity of Hepatitis C Virus Envelope Glycoprotein Complex by High-Throughput Mutagenesis." *PLoS Pathogens* 13 (12): 1–27. <https://doi.org/10.1371/journal.ppat.1006735>.
- Guan, Mo, Wenbo Wang, Xiaoqing Liu, Yimin Tong, Yuan Liu, Hao Ren, Shiyong Zhu, et al. 2012. "Three Different Functional Microdomains in the Hepatitis C Virus Hypervariable Region 1 (HVR1) Mediate Entry and Immune Evasion." *Journal of Biological Chemistry* 287 (42): 35631–45. <https://doi.org/10.1074/jbc.M112.382341>.
- Keck, Z.-Y., Sophia H Li, Jinming Xia, Thomas von Hahn, Peter Balfe, Jane A McKeating, Jeroen Witteveldt, et al. 2009. "Mutations in Hepatitis C Virus E2 Located Outside the CD81 Binding Sites Lead to Escape from Broadly Neutralizing Antibodies but Compromise Virus Infectivity." *Journal of Virology* 83 (12): 6149–60. <https://doi.org/10.1128/JVI.00248-09>.
- Keck, Zhen-yong, Jinming Xia, Yong Wang, Wenyan Wang, Thomas Krey, Jannick Prentoe, Thomas Carlsen, et al. 2012. "Human Monoclonal Antibodies to a Novel Cluster of Conformational Epitopes on HCV E2 with Resistance to Neutralization Escape in a Genotype 2a Isolate." Edited by Michael S. Diamond. *PLoS Pathogens* 8 (4): e1002653. <https://doi.org/10.1371/journal.ppat.1002653>.
- Pierce, Brian G, Zhen-Yong Keck, Patrick Lau, Catherine Fauvelle, Ragul Gowthaman, Thomas F Baumert, Thomas R Fuerst, Roy A Mariuzza, and Steven K H Fong. 2016. "Global Mapping of Antibody Recognition of the Hepatitis C Virus E2 Glycoprotein: Implications for Vaccine Design." *Proceedings of the National Academy of Sciences* 113 (45): E6946–54. <https://doi.org/10.1073/pnas.1614942113>.
- Rothwangl, Katharina B, Balaji Manicassamy, Susan L Uprichard, and Lijun Rong. 2008. "Dissecting the Role of Putative CD81 Binding Regions of E2 in Mediating HCV Entry: Putative CD81 Binding Region 1 Is Not Involved in CD81 Binding." *Virology Journal* 5 (1): 46. <https://doi.org/10.1186/1743-422X-5-46>.
- Yang, Jae Seong, Mireia Garriga-Canut, Nele Link, Carlo Carolis, Katrina Broadbent, Violeta Beltran-Sastre, Luis Serrano, and Sebastian P. Maurer. 2018. "Rec-YnH Enables Simultaneous Many-by-Many Detection of Direct Protein–Protein and Protein–RNA Interactions." *Nature Communications* 9 (1). <https://doi.org/10.1038/s41467-018-06128-x>.

Reviewers' Comments:

Reviewer #2:

Remarks to the Author:

I am happy that the authors have addressed my main concerns in this draft and thank them for taking time to address them.

I particularly, like the estimate of 100 generations as a threshold for escape time to be significant. It might be worth however emphasising this does not mean an escape time greater than 100 guarantees the site is free of escape mutants - a practical use might involve using some factor larger to give some percentage level of confidence.

Finally, I'd like to apologise since it is clear to me now that the average escape time is appropriate and overall with the ROC analysis indicates it is reasonably good metric. With the DE metric since the time was proportional to the exponential of DE, this means the minimum DE is most important. Here we are dealing directly with the escape time and so by definition those paths that give a time much smaller than the average (up to the width of the distribution) are very rare! I would suggest saying this more directly, or just removing lines 393 to 406, since currently the discussion of this is confusing.

RESPONSE TO REVIEWERS' COMMENTS

Identifying immunologically-vulnerable regions of the HCV E2 glycoprotein and broadly neutralizing antibodies that target them (NCOMMS-18-11143)

Reviewer #2 (Remarks to the Author):

I am happy that the authors have addressed my main concerns in this draft and thank them for taking time to address them.

Reciprocally, we thank the reviewer once again for the time he/she has devoted to provide detailed and thoughtful reviews. This has helped to improve our paper substantially.

I particularly, like the estimate of 100 generations as a threshold for escape time to be significant. It might be worth however emphasising this does not mean an escape time greater than 100 guarantees the site is free of escape mutants - a practical use might involve using some factor larger to give some percentage level of confidence.

We agree and have added the following text to the Results section:

The choice of $\zeta \sim 100$ is a statistically reasonable cut-off; though, if one wished to further decrease the chance of incorrectly classifying a residue as difficult to escape, a larger value of ζ could also be considered.

The point about the chosen cut-off of 100 not guaranteeing the preclusion of shorter escape times is also made in the Discussion section; please refer to the following response.

Finally, I'd like to apologise since it is clear to me now that the average escape time is appropriate and overall with the ROC analysis indicates it is reasonably good metric. With the DE metric since the time was proportional to the exponential of DE, this means the minimum DE is most important. Here we are dealing directly with the escape time and so by definition those paths that give a time much smaller than the average (up to the width of the distribution) are very rare! I would suggest saying this more directly, or just removing lines 393 to 406, since currently the discussion of this is confusing.

We completely agree and are thankful to the reviewer for giving us the opportunity to explain these concepts, which in the process allowed us to explore in detail the nuances of the average escape time metric. We have modified lines 393 to 406 as follows.

For each E2 residue, the escape time was obtained by averaging over the time it takes to observe escape in a fixed number of transmitted/founder (T/F) sequences (see Methods). We choose this average prediction measure in order to quantify the typical time for escape from an antibody targeting a specific residue. For a given residue, a higher value of the proposed escape time metric is indicative that antibody pressure directed towards that residue is generally more difficult to escape. We emphasize however that a high value of this metric does not preclude the existence of specific evolutionary scenarios (e.g., specific trajectories originating from specific T/F sequences) for which the escape time is fairly short, but these are expected to occur with very low probability.